# Mitigating Relative Over-Generalization in Multi-Agent Reinforcement Learning

**Ting Zhu[1], Yue Jin[2], Jeremie Houssineau [3], Giovanni Montana[1,2,4]**

[1]*Department of Statistics, University of Warwick, Coventry, UK*
[2]*Warwick Manufacturing Group, University of Warwick, Coventry, UK*
[3]*School of Physical & Mathematical Sciences, Nanyang Technological University, Singapore*
[4]*Alan Turing Institute, London, UK*

*ting.zhu@warwick.ac.uk*
*yue.jin.3@warwick.ac.uk*
*jeremie.houssineau@ntu.edu.sg*
*g.montana@warwick.ac.uk*

**Reviewed on OpenReview:** *https://openreview.net/forum?id=oAkSRhl3qU*

## Abstract

In decentralized multi-agent reinforcement learning, agents learning in isolation can lead to relative over-generalization (RO), where optimal joint actions are undervalued in favor of suboptimal ones. This hinders effective coordination in cooperative tasks, as agents tend to choose actions that are individually rational but collectively suboptimal. To address this issue, we introduce MaxMax Q-Learning (MMQ), which employs an iterative process of sampling and evaluating potential next states, selecting those with maximal Q-values for learning. This approach refines approximations of ideal state transitions, aligning more closely with the optimal joint policy of collaborating agents. We provide theoretical analysis supporting MMQ's potential and present empirical evaluations across various environments susceptible to RO. Our results demonstrate that MMQ frequently outperforms existing baselines, exhibiting enhanced convergence and sample efficiency.

## 1 Introduction

Cooperative multi-agent reinforcement learning (MARL) has become increasingly important for addressing complex real-world challenges that require coordinated behaviors among multiple agents. Successful applications included playing card games (Brown & Sandholm, 2018), autonomous driving (Shalev-Shwartz et al., 2016; Zhou et al., 2021), unmanned aerial vehicles (Wang et al., 2020), wireless sensor networks (Xu et al., 2020; Sahraoui et al., 2021) and traffic light control (Bazzan, 2009; Zhou et al., 2023). A dominant framework in MARL is centralized training with decentralized execution (CTDE) (Lowe et al., 2017; Rashid et al., 2018; Son et al., 2019), which often relies on a centralized coordinator to aggregate information, such as local observation or individual actions from other agents during training. While this approach has been widely adopted due to its effectiveness in leveraging global information, it may still face scalability challenges, particularly in environments with large number of agents or complex interactions. Additionally, in scenarios where privacy is a critical concern, CTDE methods that require access to individual agent data could be less desirable. While many CTDE methods do not require exhaustive local information from all agents, the potential for scalability and privacy issues in certain implementations warrants consideration. While inter-agent communication can partially mitigate some of these challenges (Foerster et al., 2016; Zhu et al., 2022), it introduces additional overhead, which can be prohibitive in environments where communication is costly or unreliable.

Consider large-scale drone swarms for search and rescue or surveillance tasks (Baldazo et al., 2019; Batra et al., 2022). In a CTDE framework, centrally aggregating data from each drone can be impractical due to network delays, dynamic environments, and potential communication failures. Additionally, communication frameworks may struggle with channel congestion or high packet loss, especially in complex terrain or with electronic interference. Fully decentralized learning presents a promising alternative, where agents rely solely on their experiences without considering the actions or policies of other agents during both training and execution. This approach enables the system to scale effectively and remain resilient to communication issues. However, decentralized approaches come with their own challenges. From an individual agent's perspective, the learning process occurs within a non-stationary MDP, as the transition probabilities change due to the evolving policies of other agents.

Existing decentralized MARL methods, including optimism strategies in Q-learning (Lauer & Riedmiller, 2000; Matignon et al., 2007; Wei & Luke, 2016), attempt to mitigate these challenges. More recently, Ideal Independent Q-learning (I2Q) (Jiang & Lu, 2022) explicitly models state transitions assuming optimal joint behavior, introducing ideal transition probabilities to address non-stationarity in independent Q-learning.

Another critical issue in decentralised MARL is *Relative Over-generalisation* (RO), where agents prefer suboptimal policies because individual actions appear preferable without coordinated strategies. First defined and studied within the field of cooperative co-evolutionary algorithms (Wiegand, 2004; Panait et al., 2006; Panait, 2007), it has more recently been explored predominantly in centralised MARL contexts (Rashid et al., 2020; Gupta et al., 2021; Shi & Peng, 2022). RO occurs when agents adapt their actions to limited interactions, often focusing on the exploratory behaviours of other agents. Consequently, agents may favour more robust but less optimal solutions in the absence of coordinated strategies. In decentralised learning settings, this problem has been discussed within simple matrix game scenarios (Wei & Luke, 2016). RO, exacerbated by non-stationarity, presents a significant challenge as agents make decisions based on fluctuating global rewards without the benefit of coordinated strategies (Matignon et al., 2012; Wei & Luke, 2016). Although previous implementations of optimistic strategies have shown some efficacy, our empirical results indicate they fall short in cooperative tasks with pronounced RO challenges.

This paper introduces MaxMax Q-Learning (MMQ), a novel algorithm designed to address the RO problem in decentralised MARL settings. MMQ aims to mitigate the challenges posed by RO and non-stationarity in decentralised learning environments. The key insight behind MMQ is enabling agents to reason about beneficial experiences that occur infrequently. At its core, MMQ employs two non-parameterised quantile models to capture the range of state transitions, accounting for both environmental factors and the evolving policies of learning agents. These models iteratively sample, evaluate, and select optimal states, refining the approximation of ideal state transitions and facilitating global reward maximisation. The state with the highest Q-value is then selected to update the value function, promoting convergence towards optimal Q-values in the context of other agents' best actions. The MMQ algorithm incorporates two maximum operators in the Bellman update: the first takes the maximum over all possible next states to select the most promising future scenario, and the second takes the maximum over Q-values of state-action pairs to determine the best action in that scenario. MMQ's adaptive nature, which involves continuously updating the ranges of possible next states, enables effective decision-making in dynamic environments.

The main contributions of this paper are threefold. First, we introduce MMQ, a novel algorithm that employs quantile models to capture multi-agent dynamics and approximate ideal state transitions through sampling. Second, we provide a theoretical demonstration of MMQ's potential to converge to globally optimal joint policies, assuming perfect knowledge of forward and value functions. Third, we present empirical results showing that MMQ often outperforms or matches established baselines across various cooperative tasks, highlighting its potential for faster convergence, enhanced sample efficiency, and improved reward maximisation in decentralised learning environments.

The remainder of this paper is structured as follows: Section 2 discusses related work in MARL and uncertainty quantification. Section 3 provides background on multi-agent Markov Decision Processes and the challenges of relative over-generalization. Section 4 presents the methodology of MaxMax Q-Learning, including its theoretical foundations and implementation details. Section 5 describes our experimental setup and results

across various environments. Finally, Section 6 concludes the paper with a discussion of our findings and potential directions for future research.

## 2 Related work

**Centralised learning methods.** Within the centralized training paradigm, RO has been discussed mostly for value factorization methods like QMIX (Rashid et al., 2018). The monotonic factorization in QMIX cannot represent the dependency of one agent's value on others' policies, making it prone to RO. Proposed solutions include weighting schemes during learning (Rashid et al., 2020), curriculum transfer from simpler tasks (Gupta et al., 2021; Shi & Peng, 2022), and sequential execution policy (Liu et al., 2024). Soft Q-learning extensions to multi-agent actor-critics (Wei et al., 2018; Lowe et al., 2017) utilise energy policies for global search to mitigate RO. However, unlike our decentralized approach, these methods require a centralized critic with joint action access during training.

**Fully decentralized learning.** Decentralized approaches in MARL aim to overcome the scalability and privacy issues associated with centralized methods. However, they face unique challenges, particularly in addressing non-stationarity and RO. Existing approaches can be categorized based on their strategies for tackling these issues. Basic independent learning methods like Independent Q-learning (IQL) (Tan, 1993) and independent PPO (Yu et al., 2022) form the foundation of decentralized MARL. However, their simultaneous updates can lead to non-stationarity, potentially compromising convergence. Recent work by Su & Lu (2023) on DPO addresses this by providing monotonic improvement and convergence guarantees. To mitigate negative impacts of uncoordinated learning, several methods promote optimism toward other agents' behaviors. Distributed Q-learning (Lauer & Riedmiller, 2000) selectively updates Q-values based only on positive TD errors. Hysteretic Q-learning (Matignon et al., 2007) uses asymmetric learning rates, while Lenient Q-learning (Wei & Luke, 2016) selectively ignores negative TD errors. These techniques aim to overcome convergence to suboptimal joint actions by dampening unhelpful Q-value changes. Taking a different approach, the recently introduced Ideal Independent Q-learning (I2Q) (Jiang & Lu, 2022) explicitly models ideal cooperative transitions. However, it requires learning an additional utility function over state pairs. Our proposed method, MMQ, builds upon these approaches by encoding uncertainty about decentralized MARL dynamics. We model other agents as sources of heteroscedastic uncertainty with an epistemic flavor, providing a more flexible way to represent optimistic policies. By sampling from possible next states, MMQ avoids the need for heuristic corrections or separate Q-functions, offering a novel solution to the challenges of decentralized MARL.

**Uncertainty quantification.** Quantifying different sources of uncertainty is crucial in reinforcement learning, particularly in multi-agent settings. Prior work distinguishes between aleatoric uncertainty, arising from environment stochasticity, and epistemic uncertainty, due to insufficient experiences (Osband et al., 2016; Depeweg et al., 2016). Various methods, including variance networks (Kendall & Gal, 2017; Wu et al., 2021) and ensembles (Lakshminarayanan et al., 2017), have been proposed to model these uncertainties, with applications in single-agent RL (Chua et al., 2018; Sekar et al., 2020). MARL introduces additional complexity due to the dynamic nature of agent interactions, leading to non-stationarity. This non-stationarity limits an agent's ability to reduce epistemic uncertainty through repeated state visits (Hernandez-Leal et al., 2017) and can be viewed as another form of epistemic uncertainty. Our proposed MMQ algorithm addresses these challenges by using quantile networks to effectively manage two key sources of epistemic uncertainty in multi-agent settings: limited experiential data and evolving strategies of other agents. This approach allows MMQ to better handle the unique uncertainties present in decentralized MARL environments.

## 3 Background and preliminaries

### 3.1 Multi-agent Markov Decision Process

Consider a multi-agent Markov Decision Process (MDP) represented by $\mathcal{M} = (\mathcal{S}, \mathcal{A}, R, P_{\text{env}}, \gamma)$. Within this tuple, $\mathcal{S}$ denotes the state space, $\mathcal{A}$ is the joint action space, $P_{\text{env}}(s'|s, \boldsymbol{a})$ and $R(s, s')$ are respectively the environment dynamics and reward function for states $s, s' \in \mathcal{S}$ and action $\boldsymbol{a} \in \mathcal{A}$, and $\gamma$ is the discount factor. Given $N$ agents, the action space is of the form $\mathcal{A} = \mathcal{A}_1 \times \cdots \times \mathcal{A}_N$ with any action $\boldsymbol{a} \in \mathcal{A}$ taking the form

Table 1: Payoff matrix for a two-agent game

|  |  | Agent 2 | | |
|---|---|---|---|---|
|  |  | A | B | C |
| Agent 1 | A | +3 | -6 | -6 |
|  | B | -6 | 0 | 0 |
|  | C | -6 | 0 | 0 |

$\boldsymbol{a} = (a_1, \ldots, a_N)$. At each time step $t$, an agent indexed by $i \in \{1, \ldots, N\}$ selects an individual action, $a_i$. When the $N$ actions are executed, the environment transitions from state $s$ to state $s'$, and every agent receives a global reward, $r_t$. The objective is to maximize the expected return, i.e., $\mathbb{E}[\sum_{t=0}^{T} \gamma^t r_t]$, where $T$ is the time horizon. The individual environment dynamics is defined as

$$P_i(s'|s, a_i) = \sum_{\boldsymbol{a}_{-i}} P_{\text{env}}(s'|s, \boldsymbol{a}) \pi_{-i}(\boldsymbol{a}_{-i}|s)$$

where $\boldsymbol{a}_{-i}$ represents the joint action excluding agent $i$ and $\pi_{-i}$ is the joint policy of all other agents. Here, the joint action $\boldsymbol{a}$ inherently depends on $a_i$ and $\boldsymbol{a}_{-i}$. From any individual agent's perspective, the learning process occurs within a non-stationary MDP due to the evolving policy $\pi_{-i}$.

### 3.2 Relative over-generalization through an example

Consider a two-agent game with the reward structure shown in Table 1. In this game, there are three possible actions: $A, B$, and $C$. Agents would receive a joint reward of $+3$ if they take $A$ together. However, if only one agent takes $A$, that agent incurs a penalty of $-6$. Agents end up choosing less optimal actions ($B$ or $C$) if they perceive the reward for choosing $A$ to be lower, based on their expectations of the other agent's actions. For agent 1, the utility function $Q_1(\cdot)$ is related to the probability that the other agent chooses $A$, i.e. $\pi_2(A)$. In this case, $Q_1(A)$ would be smaller than $Q_1(B)$ or $Q_1(C)$ if $\pi_2(A) < \frac{2}{5}$. This threshold arises because the expected value of choosing $A$ becomes lower than choosing $B$ or $C$ when the probability of the other agent also choosing $A$ falls below $\frac{2}{5}$. Consequently, during initial uniform exploration where $\pi_1(A) = \pi_2(A) = \frac{1}{3}$, both agents tend to favour $B$ or $C$ over $A$, even though $A$ is the globally optimal choice. Thus, with independent learning without considering the other agent's best action, both agents may end up with choosing suboptimal actions and fail to cooperate.

### 3.3 Independent Q-Learning

In independent Q-learning (Tan, 1993), each agent $i$ learns a policy independently, treating other agents as part of the environment. The individual Q-function is $Q_i(s, a_i) = \mathbb{E}\left[\sum_{t=0}^{\infty} \gamma^t r_t \mid s_0 = s, a_{i,0} = a_i\right]$ for agent $i$. Each agent updates its Q-function by minimizing the loss $\mathbb{E}_{P_i(s'|s,a_i)}\left[(y_i - Q_i(s, a_i))^2\right]$, where $y_i$ is the target value defined as $R(s, s') + \gamma \max_{a'_i} Q_i(s', a'_i)$. The RO problem arises in this setup as each agent seeks to maximize its own expected return based on experiences where other agents' policies evolve and contain random explorations.

### 3.4 Ideal transition probabilities

To address the RO problem in this context, some approaches introduce implicit coordination mechanisms centered on the concept of an *ideal transition model* (Lauer & Riedmiller, 2000; Matignon et al., 2007; Wei & Luke, 2016; Palmer et al., 2018; Jiang & Lu, 2022). These methods guide each agent's learning with hypothetical transitions that assume optimal joint behavior, aligning independent learners towards coordination. Let $\boldsymbol{\pi}_{-i}$ denote the joint policy of other agents, and $Q^*$ the optimal joint Q-function. The optimal joint policy of other agents can be expressed as $\pi^*_{-i}(s, a_i) = \arg\max_{\boldsymbol{a}_{-i}} Q^*(s, a_i, \boldsymbol{a}_{-i})$.

The concept of ideal transitions refers to hypothetical state transitions that assume other agents are following optimal joint policies. These ideal transition probabilities represent the dynamics that would occur if all agents achieved perfect coordination, and are defined as $P^*_i(s'|s, a_i) = P_{\text{env}}(s'|s, a_i, \pi^*_{-i}(s, a_i))$. Based on

these probabilities, the Bellman optimality equation is given by

$$Q_i^*(s, a_i) = \mathbb{E}_{P_i^*(s'|s,a_i)} \left[ R(s, s') + \gamma \max_{a_i'} Q_i^*(s', a_i') \right], \tag{1}$$

where $Q_i^*(s, a_i)$ is the optimal Q-function for agent $i$.

An important theoretical result from Jiang & Lu (2022) establishes that when all agents perform Q-learning based on these ideal transition probabilities, the individual and joint optimality align, that is, $\max_{a_i} Q_i^*(s, a_i) = Q^*(s, \pi^*(s))$, where $Q^*(s, \boldsymbol{a})$ is the optimal joint Q-function for any action $\boldsymbol{a} \in \mathcal{A}$, and $\pi^*$ is the optimal joint policy. However, achieving true ideal transitions is intractable in practice due to the evolving, uncontrolled nature of learning agents. This motivates developing techniques to approximate ideal transitions.

## 4 MaxMax Q-learning Methodology

### 4.1 Approximation of Bellman optimality equation

Our methodology aims to approximate ideal transition probabilities, which assume other agents follow optimal joint policies. We focus on deterministic environments and reformulate the Bellman optimality in Eq. (1) to highlight the dependence on the set $\mathcal{S}_{s,a_i}$ of possible next states,

$$\mathcal{S}_{s,a_i} = \left\{ s' = f_{\text{env}}(s, a_i, \boldsymbol{a}_{-i}) \mid \boldsymbol{a}_{-i} \in \prod_{j \neq i} \mathcal{A}_j \right\}, \tag{2}$$

where $f_{\text{env}}(s, a_i, \boldsymbol{a}_{-i})$ is the deterministic transition function that maps the current state $s$ and the joint actions of all agents $(a_i, \boldsymbol{a}_{-i})$ to the next state $s'$, $\mathcal{A}_j$ is the action space for each agent $j$ and the Cartesian product $\prod_{j \neq i} \mathcal{A}_j$ represents all possible combinations of actions by the other agents. It is noted that we only use $f_{\text{env}}$ to define the set $\mathcal{S}_{s,a_i}$ here, but do not need to learn this global transition function directly in our algorithm. Encoding the deterministic transitions by a delta function, $\delta_{f_{\text{env}}(s,\boldsymbol{a})}(s')$, Eq. (1) is rewritten as

$$Q_i^*(s, a_i) = \mathbb{E}_{s' \sim \delta_{f_{\text{env}}(s,a_i,\pi_{-i}^*(s,a_i))}(s')} \left[ R(s, s') + \gamma \max_{a_i'} Q_i^*(s', a_i') \right] \tag{3a}$$

$$= \max_{s' \in \mathcal{S}_{s,a_i}^*} \left( R(s, s') + \gamma \max_{a_i'} Q_i^*(s', a_i') \right). \tag{3b}$$

where $\mathcal{S}_{s,a_i}^*$ is any subset of $\mathcal{S}_{s,a_i}$ including $s'^* = f_{\text{env}}(s, a_i, \pi_{-i}^*(s, a_i))$.

By reformulating the Q-value optimization over the set $\mathcal{S}_{s,a_i}^*$, our approach allows for targeting the optimal value $Q^*(s, a_i)$ under the true coordinated joint behavior, without directly approximating $s'^*$. When there is no information about $s'^*$, the set $\mathcal{S}_{s,a_i}^*$ could be set in principle to $\mathcal{S}_{s,a_i}$, if it were known, to ensure the inclusion of $s'^*$. However, this will also make the maximisation over $\mathcal{S}_{s,a_i}^*$ in Eq. (3b) more computationally challenging, implying a trade off between reducing the size of $\mathcal{S}_{s,a_i}^*$ and ensuring the inclusion of $s'^*$. We propose a learning procedure that enables each agent to progressively shrink their set of next states $\mathcal{S}_{s,a_i}^*$, as all agents explore and accumulate experiences.

In practice, at the algorithmic step $t$, we work with a subset $\hat{\mathcal{S}}_{s,a_i,t}$ which approximates one of the possible subsets $\mathcal{S}_{s,a_i}^*$. Since neither $\mathcal{S}_{s,a_i}$ nor $s'^*$ are known in practice due to the incomplete information about other agents' policies and the environment dynamics, we cannot guarantee that $s'^* \in \hat{\mathcal{S}}_{s,a_i,t} \subseteq \mathcal{S}_{s,a_i}$ holds, but we will show in our performance assessment that $s'^* \in \hat{\mathcal{S}}_{s,a_i,t}$ holds with high probability. Furthermore, direct maximization over $\hat{\mathcal{S}}_{s,a_i,t}$ is challenging as this set is infinite in general.

To address this, we resort to Monte Carlo optimization, as in e.g. Robert et al. (1999), by introducing a finite set $\hat{\mathcal{S}}_{s,a_i,t}^M$ of $M$ points randomly sampled from $\hat{\mathcal{S}}_{s,a_i,t}$. Assuming no approximation error in the predicted

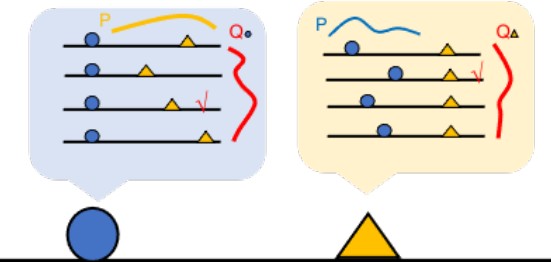

Figure 1: Illustration of the MMQ update for two agents. Different positions of two agents in the rounded rectangles represent different possible next states $s' = (x_b, x_y)$. From the perspective of the blue agent: The yellow curve (**P**) represents the distribution of states with different yellow agent positions ($x_y$) in the replay buffer. The red curve (**Q**) represents the estimated Q-values for those possible next states. In the MMQ update, the blue agent selects samples for update based on the highest Q-value, marked by $\sqrt{}$. Importantly, this selection may not always coincide with the most frequently encountered scenarios (corresponding to the peak of the yellow curve) from past experiences, which may be sub-optimal.

bound, that is $\hat{\mathcal{S}}_{s,a_i,t} \subseteq \mathcal{S}_{s,a_i}$ holds, the sample set $\hat{\mathcal{S}}^M_{s,a_i,t}$ contains only reachable states and it follows that

$$Q_i^*(s, a_i) \geq \max_{s' \in \hat{\mathcal{S}}^M_{s,a_i,t}} \left[ R(s, s') + \gamma \max_{a_i'} Q_i^*(s', a_i') \right] \tag{4a}$$

$$= \max_{m \in \{1,...,M\}} \left[ R(s, s_m') + \gamma \max_{a_i'} Q_i^*(s_m', a_i') \right]. \tag{4b}$$

With the considered approach, there are two natural phases when running the associated algorithms:

1. With little information to rely on, the agents explore the state space at random and collect diverse trajectories, which improve their understanding of the range of possible next state $\mathcal{S}_{s,a_i}$. In this phase, the estimated sets $\hat{\mathcal{S}}_{s,a_i,t}$ will be close to $\mathcal{S}_{s,a_i}$.
2. As agents refine their estimates of the set $\mathcal{S}_{s,a_i}$ of possible next states and accumulate sufficient reward information, the maximisation in Eq. (4b) yields increasingly stable values, facilitating policy convergence. This, in turns, means that the new trajectories will be more similar and optimised, progressively outnumbering the initial diverse trajectories. This will cause the sets $\hat{\mathcal{S}}_{s,a_i,t}$ to zero in on $s'^*$, hence facilitating the Monte Carlo optimisation in Eq. (4b).

An illustration of our sampling and selection process is shown in Figure 1: given a set of possible next state samples, our algorithm selects the state with the highest estimated Q-value for updating, which implicitly indicating the optimal action of other agents. As agents explore more possible actions, the estimated set $\hat{\mathcal{S}}_{s,a_i,t}$ increasingly approximates the true set $\mathcal{S}_{s,a_i}$. Crucially, if the optimal next state is contained within the estimated set, the equality

$$Q_i^*(s, a_i) = \max_{s' \in \hat{\mathcal{S}}_{s,a_i,t}} \left[ R(s, s') + \gamma \max_{a_i'} Q_i^*(s', a_i') \right]$$

holds. This property is fundamental to our method, as it implies that through iterative learning and effective sampling, each agent can learn Q-values that closely align with those derived from ideal transition probabilities. In the following section, we analyse the convergence properties of this approach under ideal conditions. The complete algorithm, including implementation details, will be presented in Section 4.3.

## 4.2 Convergence analysis

Our combined learning and sampling approach facilitates the gradual convergence of the agents' policies toward the globally-optimal joint policy. This gradual convergence is supported by the insights from the following theorem, which shows the disparity between the optimal Q-values and those learned by the agents

is limited by the difference between the best next state in the estimated set and the true best next state. This bounding relationship is crucial, as it indicates that the closer our estimated set of next states is to the actual set composed of all the possible next states, the more accurate the estimations of the agents' optimal Q-functions become.

In this section, we further elaborate on this mechanism and provide a formal convergence analysis. This analysis demonstrates how our proposed model-based Q-learning approach, combined with the sampling strategy, effectively facilitates convergence to an optimal global policy. We begin by showing that the difference between the Q-values learned by our approach and the optimal Q-values depends on how well we can estimate the best next state.

**Theorem 4.1.** *Let $\mathcal{S}_{s,a_i}$ be the set of all possible next states as defined in* (2) *and let $\hat{S}$ be a non-empty subset of $\mathcal{S}_{s,a_i}$. Let $s'^*$ and $\hat{s}'^*$ represent the best next states in the optimal and approximate regimes, respectively, that is*

$$s'^* = \arg\max_{s' \in \mathcal{S}_{s,a_i}} R(s, s') + \gamma \max_{a_i'} Q_i^*(s', a_i')$$

$$\hat{s}'^* = \arg\max_{s' \in \hat{S}} R(s, s') + \gamma \max_{a_i'} Q_i(s', a_i').$$

*Under Assumptions A.1-A.3 (see Appendix B), if the Euclidean distance $d(s'^*, \hat{s}'^*)$ is at most $\epsilon$ for all $(s, a_i)$, then there exists $K > 0$ such that $|Q_i^*(s, a_i) - Q_i(s, a_i)| \leq (1 - \gamma)^{-1} K \epsilon$ for all $(s, a_i)$.*

The proof can be found in Appendix B.

In the context of our algorithm, we can relate this theorem to our specific implementation. Omitting the algorithm step $t$ from the notations for simplicity, $\hat{S}$ in our case corresponds to $\hat{\mathcal{S}}_{s,a_i}^M$. This set consists of $M$ states uniformly sampled from $\hat{\mathcal{S}}s, a_i$, which is itself a non-empty subset of $\mathcal{S}s, a_i$. $\hat{\mathcal{S}}_{s,a_i}$ is formed by all possible outcomes predicted by our learned model. Figure 2 illustrates the relationships between these sets.

This result demonstrates that if the distance between the estimated best next state $\hat{s}'^*$ and the actual best next state $s'^*$ is arbitrarily small for all state-action pairs $(s, a_i)$, then the discrepancy between the learned Q-values $Q_i(s, a_i)$ and the optimal Q-values $Q_i^*(s, a_i)$ is bounded. This implies that as we refine our estimation of the optimal next state through iterative sampling and learning, we progressively narrow the gap between the learned Q-values of our agents and the true optimal values.

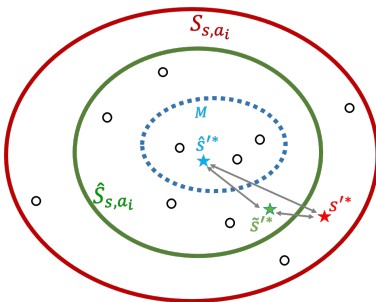

Figure 2: Illustration of the set relationship among $\mathcal{S}_{s,a_i}$, $\hat{\mathcal{S}}_{s,a_i}$ and $\hat{\mathcal{S}}_{s,a_i}^M$ (denoted as $M$ above). The red star, $s'^*$, is the best next state in the real set, and $\tilde{s}'^*$, and $\hat{s}'^*$ represent the two states that are the closest to the best next states in $\hat{\mathcal{S}}_{s,a_i}$ and $\hat{\mathcal{S}}_{s,a_i}^M$. According to the triangle inequality, the distance between $s'^*$ and $\hat{s}'^*$, $d(s'^*, \hat{s}'^*)$, is upper bound by the sum of $d(s'^*, \tilde{s}'^*)$ and $d(\tilde{s}'^*, \hat{s}'^*)$

To better analyse the distance $d(s', \hat{s}')$, we introduce a third state $\tilde{s}'$ as the best next state in $\hat{\mathcal{S}}_{s,a_i}$. This allows us to use the triangle inequality to upper bound the distance as , $d(s', \hat{s}') \leq d(s', \tilde{s}') + d(\tilde{s}', \hat{s}')$

The first term, $d(s', \tilde{s}')$, reflects the difference between the optimal next state in $\mathcal{S}_{s,a_i}$ and the one in $\hat{\mathcal{S}}_{s,a_i}$. As the size of $\hat{\mathcal{S}}_{s,a_i}$ increases in the first phase of the algorithm, it's more likely to include states closer to $s'$, thus decreasing $d(s', \tilde{s}')$. The second term, $d(\tilde{s}', \hat{s}')$, represents the error in the Monte Carlo optimization. This error tends to be large initially but decreases in the second phase as $\hat{\mathcal{S}}_{s,a_i}$ shrinks, making it easier to

sample states close to $\tilde{s}'$. This analysis shows how our algorithm progressively improves its estimation of the optimal next state, contributing to the overall convergence of the Q-values.

**Theorem 4.2.** *Assume that $\mathcal{S} = \mathbb{R}$ and that $\hat{\mathcal{S}}_{s,a_i}$ is of the form $[-u, u]$ for some $u \in (0, \infty)$. Consider $M$ i.i.d. samples $s_1', \ldots, s_M'$ from the uniform distribution on $[-u, u]$. It holds that*

$$\mathbb{E}\left[ \min_{k=1,\ldots,M} |\tilde{s}'^* - s_k'| \right] < \frac{2u}{M+1}.$$

The proof can be found in Appendix B.

This result demonstrates that the Monte Carlo optimization error $d(\tilde{s}'^*, \hat{s}'^*)$ diminishes as the number of samples $M$ increases. Incidentally, the Monte Carlo optimisation error could exhibit exponential dependence as the dimensionality of the state space increases. However, in multi-agent scenarios, partially-observed MDPs typically limit effective dimensionality growth, as agents rely on a restricted view of the overall state space.

Seemingly, there is an inherent trade-off involved in expanding the state set $\hat{\mathcal{S}}_{s,a_i}$, i.e., expanding $u$ in the above one-dimension case, to cover more possibilities while managing the resultant error. A broader $\hat{\mathcal{S}}_{s,a_i}$ reduces the gap between $\tilde{s}'^*$ and the true best state $s'^*$, as it increases the likelihood of encompassing $s'^*$. This action effectively shrinks the error term $d(s'^*, \tilde{s}'^*)$. Yet, increasing the size of $\hat{\mathcal{S}}_{s,a_i}$, i.e., increasing $u$, also typically increases variability, leading to a larger error $d(\tilde{s}'^*, \hat{s}'^*)$ and necessitating more samples to maintain a given level of precision.

### 4.3 Implementation details

---

**Algorithm 1:** MMQ for each agent $i$

---

**Input:** Q-network $Q_i$, actor network $\pi_i$ and target networks $\overline{Q}_i$, $\overline{\pi}_i$; Quantile models $g_i^{\tau_l}$ and $g_i^{\tau_u}$; Reward network $R_i$; Replay buffer $D_i$

**for** $t = 1, \ldots, T_{\max}$ **do**

    All agents interact with the environment using random action (for initial $P$ steps) or $\varepsilon$-greedy policy and store experiences $(s, a_i, r, s')$ in $D_i$

    Sample a mini-batch from $\mathcal{D}_i$;

    Update $g_i^{\tau_l}$ and $g_i^{\tau_u}$ by minimizing (5)

    Draw $M$ samples from predicted bound $[g_i^{\tau_l}(\cdot), g_i^{\tau_u}(\cdot)]$ to construct set $\hat{\mathcal{S}}$

    Calculate the target value over $\hat{\mathcal{S}}$ using (6)

    Update $Q_i$ multiple times by minimizing (7)

    Update $R_i$ by minimizing (8)

    Update $\pi_i$ by minimizing (9)

    Update the target network $\overline{Q}_i$ and $\overline{\pi}_i$

**end for**

---

To capture the range of possible next states, our implementation utilises two non-parametrised quantile models, $g_i^{\tau_l}$ and $g_i^{\tau_u}$, which employ neural networks to predict the $\tau_l = 0.05$ and $\tau_u = 0.95$ quantiles for each dimension of the next state. The neural network parameters, denoted by $\phi_i^l$ and $\phi_i^u$, are learnt by minimising the quantile loss according to their respective $\tau$ values over samples from individual replay buffer $D_i$:

$$L(\phi_i) = \mathbb{E}_{s,a_i \sim \mathcal{D}_i}[L^\tau(g_i^\tau(s, a_i; \phi_i) - s')], \tag{5}$$

where $L^\tau(u) = \mathbb{I}(u > 0)\tau u + \mathbb{I}(u < 0)(1 - \tau)u$. For each $(s, a_i)$ pair, the two quantile models predict bounds $[g_i^{\tau_l}(s, a_i), g_i^{\tau_u}(s, a_i)]$. We then construct the potential next state set $\hat{\mathcal{S}}$ by including the true $s'$ and $M$ samples drawn from the quantile bounds. We also explored a parametrised multivariate Gaussian model as another method to estimate the possible next states, detailed in Appendix C.3.

Each agent also learns a Q-network, parameterised by $\theta_i$, and represented as $Q_i(s, a_i; \theta_i)$. The target value for the Q-network update is given by:

$$Y_i(s, a_i; \theta_i) = \max_{\hat{s}' \in \hat{\mathcal{S}}} \left( R_i(s, s'; \psi_i) + \gamma \max_{a_i'} Q_i(\hat{s}', a_i'; \theta_i) \right), \tag{6}$$

where $R_i(s, s'; \psi_i)$ is an estimate from a learned reward model parameterised by $\psi_i$. Here we apply a stop-gradient operator to the target value to prevent gradient flow back to the next state estimation process. This operation is crucial for maintaining stability in the learning dynamics by separating the optimization of the Q-network from the updates to the next state estimation. The loss function for optimising the Q-network parameter $\theta_i$ is then given by

$$L(\theta_i) = \mathbb{E}_{s, a_i \sim \mathcal{D}_i} \left[ Q_i(s, a_i; \theta_i) - Y_i(s, a_i; \theta_i) \right]^2. \tag{7}$$

This loss aims to align the Q-network's predictions with the maximum expected return, considering both the immediate reward and the discounted future Q-values of the potential next states sampled from the estimated quantile bound. Additionally, the learned reward function $R_i(s, s'; \psi_i)$, parameterised by $\psi_i$, is trained to approximate the rewards for state transitions. The loss for this reward model is the mean squared error between the predicted and actual rewards,

$$L(\psi_i) = \mathbb{E}_{s \sim \mathcal{D}_i} [R_i(s, s'; \psi_i) - r]^2. \tag{8}$$

To deal with continuous action spaces, each agent uses an actor network $\pi_i(s; \rho_i)$, parameterised by $\rho_i$, to learn its policy. The loss function for the actor network, aimed at maximising the Q-value, is

$$L(\rho_i) = \mathbb{E}_{s \sim \mathcal{D}_i} [-Q_i(s, \pi_i(s; \rho_i); \theta_i)]. \tag{9}$$

The training process is summarised in Algorithm 1, and the full source code is available at `https://github.com/Tingz0/Maxmax_Q_learning`. The full algorithm interleaves the optimisation of various constituent models, allowing agents to adaptively learn and improve their policies based on their own experiences and the evolving environmental dynamics. Our implementation incorporates two key strategies. First, a delayed update approach for the actor network relative to the critic network, where the critic is updated 10 times more frequently to maintain stability (Fujimoto et al., 2018). Second, negative reward shifting (Sun et al., 2022), which enhances our double-max-style updates (see also Appendix C.1).

## 5 Experimental results

### 5.1 Environments

We evaluated the MMQ algorithm in three types of cooperative MARL environments characterized by the need for complex coordination among agents; see Figure 3 for an overview.

**Differential Games** We adapted this environment from Jiang & Lu (2022), where $N$ agents move within the range $[-1, 1]$. At each time step, an agent indexed by $i$ selects an action $a_i \in [-1, 1]$. The state of this agent then transitions to clip$\{x_i + 0.1 \times a_i, -1, 1\}$, where $x_i$ is the previous state and the clip$(y, y_{\min}, y_{\max})$ function restricts $y$ within $[y_{\min}, y_{\max}]$. The global state is the position vector $(x_1, x_2)$. The reward function, detailed in Appendix A, assigns rewards following each action. A narrow optimal reward region is centred, surrounded by a wide zero-reward area and suboptimal rewards at the edges (see DG in Figure 3). This setup can lead to RO problems as agents might prefer staying in larger suboptimal areas.

**Multiple Particle Environment** We designed six variants of cooperative navigation tasks with RO rewards as shown in Figure 3. The common goal is for two disk-shaped agents, $D_1$ and $D_2$, to simultaneously reach a disk-shaped target. To encourage coordination, we introduce a penalty for scenarios where only one agent is within a certain distance from the target. Specifically, we define a disk $D$ centered on the target with

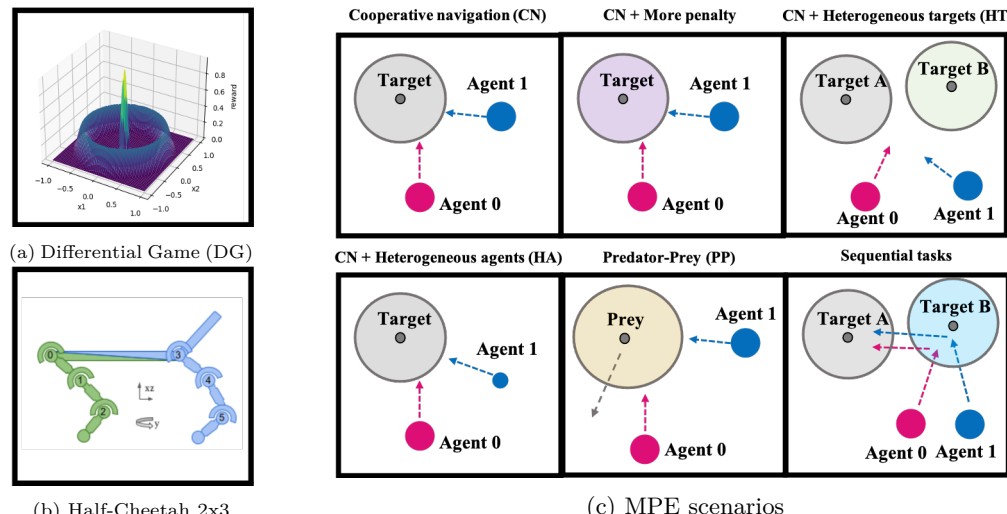

Figure 3: Task visualization. (a) *Differential Game*(DG): agents need to cross a wide zero-reward area to move to the center to gain the optimal reward. (b) *Half-Cheetah 2x3*: the Half-Cheetah 2x3 scenario in MAmujoco domain; (c) *MPE scenarios*; *Cooperative navigation*(CN): two agents need to enter the grey area of the target together to gain the reward, the solo entry would induce a penalty. *CN + More penalty*: Same task as CN but with more penalty for solo entry; *CN + HT*: Agents could choose to approach one of the two Targets with different reward settings; *CN+HA*: same task as CN but two agents have different sizes and velocity; *Predator-Prey(PP)*: two agents need to enter the grey area of a pre-trained prey. *Sequential Task*: two agents need to first go through the grey area of Target B and then enter the grey of Target A with the same RO reward design as CN.

radius $r_D$ and penalize agents if only one is within $D$. This setup is designed to illustrate the RO problem, where agents might prefer staying outside $D$ rather than risk being the only one inside it. The task difficulty increases as the radius $r_D$ decreases. The reward function, designed to reflect the RO problem, is defined as:

$$r_{\text{CN}} = \begin{cases} R_{in} & \text{if } D_i \cap D \neq \emptyset, i = 1, 2 \\ R_{out} & \text{if } (D_1 \cup D_2) \cap D = \emptyset \\ R_{out} - p & \text{otherwise.} \end{cases}$$

The rewards for the three cases should satisfy $R_{out} - p < R_{out} < R_{in}$. Detailed descriptions of $R_{out}$ and $R_{in}$ for different settings are provided in Appendix A. In task **CN**, the penalty for solo entry into the circle is $p = 0.2$; in task **CN+More Penalty**, the penalty increases to $p = 0.5$ for entering the circle alone; in task **CN+Heterogeneous Agents (HA)**, two agents performing the **CN** task are heterogeneous, having different sizes and velocities; in task **CN+Heterogeneous Targets (HT)**, there are two targets, where entering the circle of target $A$ follows the previous RO design, and entering the circle of target $B$ incurs no RO penalty but offers a reward lower than $R_{in}$; in the sub-optimal scenario, agents might only enter the circle of target $B$; in task **Sequential Task**, agents must coordinate over a longer period—they could either directly reach target $A$ with the same RO reward as before or first reach target $B$ to pick up cargo, then receive a bonus each step (a higher $R_{in}$) when they later enter target $A$ together; in task **Predator-Prey (PP)**, two predators (which we control) and one prey, who interact in an environment with two obstacles. The prey, trained using MADDPG (Lowe et al., 2017), is adept at escaping faster than the predators. The predators need to enter the prey's disk together to receive the reward $R_{in}$.

**Multi-agent MuJoCo Environment**   We employ the Half-Cheetah 2x3 scenario from the Multi-agent MuJoCo framework (de Witt et al., 2020). This environment features two agents, each controlling three joints of the Half-Cheetah robot via torque application. It presents a partial observability setting, with each agent accessing only its local observations. We implement an RO reward structure designed to necessitate

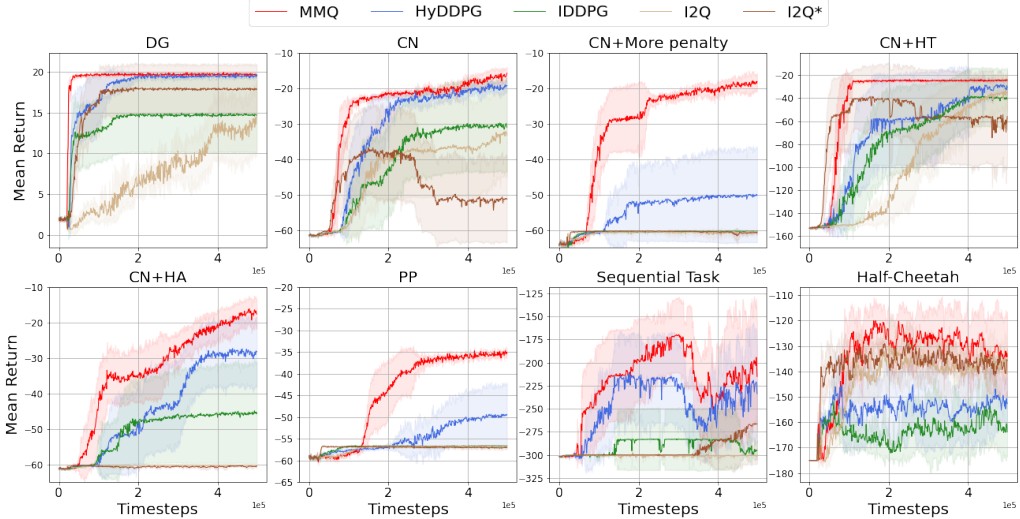

Figure 4: Performance comparison for two-agents setting in DG, MPE scenarios and Half-Cheetah

high coordination between agents. The reward function $r_v$ is defined as: $-7$ if $v < v_l$, $-9$ if $v_l \leq v \leq v_u$, and $-2$, otherwise; where $v_l = 0.035/dt$, $v_u = 0.04/dt$, and $dt = 0.05$. This structure rewards agents for moving forward at speeds exceeding $v_u$, penalizes speeds between $v_l$ and $v_u$, and provides a moderate penalty for very low speeds. Agents failing to overcome the RO problem may settle for maintaining low speeds to avoid the harshest penalty.

Table 2: Settings with $N = 2$ agents. Mean Returns and 95% Confidence Interval (over eight seeds) for all algorithms at the end of training. Values are bolded if their confidence intervals overlap with the maximum value.

| Setting | Setting | MMQ | IDDPG | HyDDPG | I2Q* | I2Q |
|---|---|---|---|---|---|---|
| DG | $N = 2$ | **19.55±0.16** | 14.67±4.61 | **19.47±0.17** | **17.84±3.01** | 14.62±4.23 |
| MPE Tasks | CN | **-15.66±1.75** | -30.91±12.78 | **-19.12±1.56** | -51.24±12.01 | -33.16±11.82 |
| | CN+more penalty | **-18.01±2.24** | -60.25±0.07 | -50.12±13.24 | -60.55±0.36 | -60.54±0.70 |
| | CN+HT | **-24.25±0.49** | **-40.34±26.01** | **-30.93±6.92** | **-56.04±40.80** | **-35.95±15.83** |
| | CN+HA | **-17.63±4.12** | -45.76±13.99 | **-28.21±9.71** | -60.49±0.15 | -60.25±0.03 |
| | PP | **-35.28±0.40** | -56.63±0.11 | -49.40±7.18 | -57.10±0.33 | -56.67±0.14 |
| | Sequential Task | **-215.09±59.97** | -295.31±9.75 | **-233.55±53.21** | -300.53±0.45 | **-266.42±43.37** |
| Half-Cheetah | $2 \times 3$ | **-134.09±16.05** | -163.81±9.61 | -152.66±10.32 | **-135.65±4.60** | **-140.94±8.55** |

Table 3: Setting with more than 2 agents. Mean Returns and 95% Confidence Interval (over eight seeds) for all algorithms at the end of training. Values are bolded if their confidence intervals overlap with the maximum value.

| Setting | Setting | MMQ | IDDPG | HyDDPG | I2Q* | I2Q |
|---|---|---|---|---|---|---|
| DG | $N = 3$ | **19.51±0.15** | 15.45±3.76 | **16.17±4.11** | 15.95±3.78 | 8.19±3.85 |
| | $N = 4$ | **20.09±0.10** | 14.09±4.22 | 12.77±4.56 | **16.31±4.28** | 12.59±4.45 |
| | $N = 5$ | **20.44±0.11** | 13.94±4.27 | 16.02±3.97 | **16.91±3.30** | 2.98±0.70 |
| MPE Tasks | CN (N=3) | **-34.79±2.75** | -60.11±17.06 | **-45.47±15.48** | **-50.86±17.68** | -54.89±17.49 |
| | PP (N=3) | **-37.86±1.01** | -49.64±2.14 | -46.16±4.84 | -52.52±0.26 | -51.86±0.95 |

## 5.2 Baselines

Our benchmarks include comparisons with three baseline algorithms: Ideal Independent Q-Learning (I2Q), Independent Deep Deterministic Policy Gradient (IDDPG), and Hysteretic DDPG (HyDDPG). We differ-

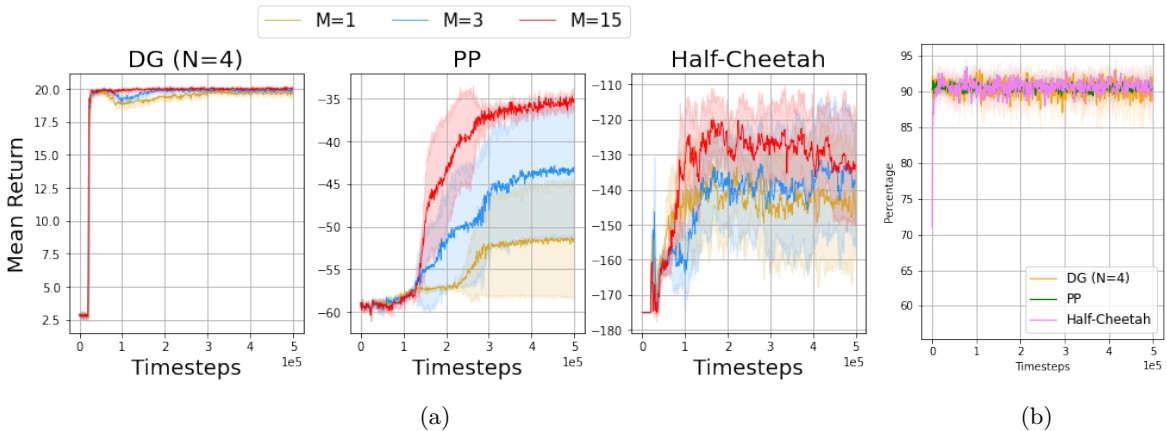

Figure 5: (a) Ablation study for different sample number $M$ in three tasks; (b) Percentage of each dim of true next states fall within the predicted quantile bound for three tasks

entiate between two versions of I2Q: the original implementation by Jiang & Lu (2022), which we refer to as I2Q*, involves multiple updates of all network components after every 50 interaction steps. In contrast, our implementation, denoted as I2Q, updates only the critic network multiple times every 50 steps. This distinction allows us to more accurately assess the performance impact of these differing update strategies.

### 5.3 Experiment Results

To compare the performance among different baselines, we consider the mean episode return over 8 seeds, where the episode return is defined as the accumulated reward $\sum_{t=0}^{T} r_t$ for the whole episode, with $T$ being the episode length.

**Differential Games** Our evaluations, depicted in Figure 4 and Table 2, show that MMQ outperforms other algorithms with 15 samples drawn from the quantile bounds predicted by two quantile models. HyDDPG and I2Q* also perform well. Interestingly, the performance of HyDDPG and IDDPG surpasses that reported for I2Q in Jiang & Lu (2022), possibly due to our implementation's emphasis on updating the critic network more frequently than the actor network to stabilize training. However, I2Q learns much slower compared to I2Q*, which updates all modules rather than just the critic multiple times. These update strategies have different effects on various algorithms. With more agents, the performance of other algorithms degrades slightly, showing higher variability, as shown in Table 3. MMQ consistently identifies the optimal region in all test cases, demonstrating its higher sample efficiency and scalability. We also tested MMQ in a stochastic version of this game (Appendix D.1), confirming that MMQ still outperforms the baselines under stochastic state transitions and reward dynamics.

**Multiple Particle Environment** As shown in Figure 4 and Table 2, our algorithm, using 15 samples from the predicted quantile bounds, successfully overcomes the Relative Over-generalisation (RO) problem in all settings. HyDDPG performs particularly well, especially in the lower penalty scenarios, such as **CN**, **CN+HT**, and **CN+HA**. In **CN** and **CN+HT**, I2Q* initially demonstrated some ability to solve the task but its performance deteriorated over time. We observed that the learned Q-values of some seeds increased rapidly and incorrectly simultaneously, despite setting the weight of the $Q^{ss}(s, s')$ value to a minimal parameter during the update process. This might be due to the challenges in computing effective $Q^{ss}(s, s')$ values in I2Q, leading to less accurate predictions of optimal states and resulting in cumulative estimation errors over time. With a higher solo-entry penalty in **CN+More Penalty**, all baselines' performance significantly declines, remaining stuck in suboptimal areas except for HyDDPG, which could learn to a limited extent. Our sampling-based approach, however, demonstrates robustness even with the increased RO problem.

**PP** presents a more challenging task compared to the above settings due to its larger state space, more agents, and fast-moving prey that the predators must catch. Our results, illustrated in Figure 4, show our algorithm successfully overcomes the RO problem, demonstrating higher mean return and greater sample efficiency than other baselines.

In the **Sequential Task**, both our MMQ and HyDDPG initially learned to enter target $A$ directly. After discovering that entering target $B$ could yield a bonus, they deliberately targeted $B$, which led to a temporary decline in performance. MMQ recovered quicker than HyDDPG. I2Q began to show learning towards the end of the training while IDDPG failed in this task.

We also tested our algorithm in the **CN** and **PP** settings with an additional agent (as shown in Table 3). With one more agent, the state dimension increases and the transition dynamics become more complex. In **CN** with three agents, the results for HyDDPG and I2Q* showed large variability, indicating that these methods were effective only under certain initial conditions. In **PP** with three agents, which is more challenging, all baselines were stuck in suboptimal areas. However, our algorithm still managed to overcome the RO problem, demonstrating scalability with an increased number of agents. Further tests with the default reward setting (detailed in Appendix D.2) show that our algorithm matches the performance of other baselines in settings without significant RO problems.

**Multi-agent MuJoCo environment**   In this setting, we also utilized 15 samples drawn from the predicted quantile bounds. As depicted in Figure 4, MMQ remains competitive in this challenging environment, consistently surpassing other baselines during the latter half of the training period. Given the inherent complexity of the task, we maintained a mild level of the RO problem to preserve feasibility. This explains why I2Q was able to perform well, despite its sensitivity to RO problems noted in other environments. We provided results for one more partitions of Half-Cheetah (Half-Cheetah 4|2) and two partitions of Ant (Ant $2 \times 4$ and $4 \times 2$) with the same reward setting in Appendix D.3. In these tasks, MMQ consistently outperforming other baselines.

**Ablation study: number of samples**   We conducted an ablation study to investigate the effect of varying the number of samples across three different environmental settings, as shown in Figure 5a. The results demonstrated that MMQ, even with just one sample, outperformed baselines in the DG and Half-Cheetah environments. In the PP environment, using one sample was slightly less effective than HyDDPG, but using three or more samples consistently outperformed all baselines. Additionally, increasing the number of samples $M$ appeared to accelerate learning across the three settings, aligning with our theoretical analysis. Furthermore, we included a study (see Appendix C.2) that employed a small ensemble of quantile models. Although this ensemble enlarged the predicted bounds and enhanced the percentage of true next states within these bounds, as shown in Figure 8b, it did not lead to further performance improvements. Thus, we did not incorporate the ensemble approach in the final results for simplicity.

To demonstrate the effectiveness of the quantile model, Figure 5b shows the percentage of each dimension of the true next states that fall within the predicted quantile bounds during the learning process for three environment settings. The percentage is calculated as follows: For each sample in the mini-batch, consisting of $n$ samples, during the update process, we have the predicted bound $[l_d, u_d]$ for each dimension $d$ of the state with value $s_d$. The state has a total of $D$ dimensions. If the state value $s_d$ falls within the predicted bound, i.e., $l_d \leq s_d \leq u_d$, we increment a count. If the total count across all dimensions and samples is $P$, the percentage is then calculated as $\frac{P}{n \cdot D} \times 100\%$.

The percentage is already high initially for the DG and CN environments but is a bit lower for Half-Cheetah, possibly due to its more complex dynamics compared to the other two. These results indicate that the quantile model can effectively capture most state changes as expected, suggesting that $\hat{\mathcal{S}}_{s,a_i}$ closely reflects the true set $\mathcal{S}_{s,a_i}$, as analyzed in Section 4.1.

## 6   Discussion and Conclusions

In this work, we introduced MaxMax Q-learning (MMQ), a novel algorithm designed to mitigate the Relative Over-generalisation problem in multi-agent collaborative tasks through the use of quantile models and

optimistic sampling. Our key theoretical contribution establishes a connection between the accuracy of estimating the best next state and the convergence towards globally optimal joint policies within the MMQ framework, providing a solid foundation for understanding the algorithm's behaviour and performance.

Empirically, we demonstrated the effectiveness of MMQ across three diverse tasks, showing its ability to outperform existing baselines in terms of convergence speed, sample efficiency, and final performance. A notable outcome is MMQ's scalability, as it can accommodate an increased number of agents while achieving superior performance with minimal samples, which reduces computational overhead—a crucial factor in practical multi-agent systems.

Despite these promising results, several limitations of MMQ remain. First, the Monte Carlo optimisation error in MMQ may exhibit exponential dependence as the dimensionality of the state space increases. However, this issue is somewhat mitigated in multi-agent settings, where the use of partially-observed MDPs limits the effective dimensionality. Second, although MMQ has demonstrated scalability in our experiments, challenges may arise in environments with very large populations of agents, where increased computational costs and coordination complexity could impact performance.

Looking ahead, future work may aim to address these limitations and further enhance MMQ's capabilities. One promising direction involves developing adaptive mechanisms to gauge the informativeness of observations about other agents, which could improve the robustness of reward function learning, especially in scenarios with partial or noisy observations. Additionally, relaxing the current assumption of independence across dimensions in the quantile model by predicting the covariance matrix could capture more complex state dynamics, potentially leading to more accurate estimations. Further improvements in scalability might be achieved by reducing the number of samples required as the state space grows, thereby lowering computational demands in high-dimensional and large-agent scenarios. Together, these future developments aim to overcome current limitations and extend the applicability of MMQ to a broader range of multi-agent scenarios.

**Acknowledgments** TZ acknowledges support from the UK Engineering and Physical Sciences Research Council (EPSRC EP/W523793/1), through the Statistics Centre for Doctoral Training at the University of Warwick. GM acknowledges support from a UKRI AI Turing Acceleration Fellowship (EPSRC EP/V024868/1).

## Broader Impact Statement

This paper presents work whose goal is to advance the field of decentralized learning of multi-agent systems. There are many potential societal consequences of our work, none of which we feel must be specifically highlighted here.

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
