## A   Environments: further details

**Differential Games.** In the differential games (Jiang & Lu, 2022), $N$ agents can move within the range $[-1, 1]$. At each time step, agent $i$ selects an action $a_i \in [-1, 1]$, and the state of agent $i$ is updated to $\text{clip}\{x_i + 0.1 \times a_i, -1, 1\}$, where $x_i$ is the previous state of agent $i$. The $\text{clip}(y, y_{\min}, y_{\max})$ function constrains $y$ to the interval $[y_{\min}, y_{\max}]$. The global state is represented by the position vector $(x_1, x_2)$. Agents receive rewards after each action, with the reward function for each time step defined as:

$$
r = \begin{cases}
a(\cos(l\pi/m) + 1) & \text{if } l \leq m \\
0 & \text{if } m < l \leq 0.6 \\
b(\cos(5\pi(l - 0.8)) + 1) & \text{if } 0.6 < l \leq 1 \\
0 & \text{if } l > 1
\end{cases}
$$

where $a$ and $b$ define the optimal and sub-optimal reward values, respectively; $m$ determines the width of the zero-reward area between the sub-optimal and optimal reward zones. We set $a = 0.5$, $b = 0.15$, and $m = 0.13(N - 1)$. The location metric, $l$, is calculated as:

$$
l = \sqrt{\frac{2}{N} \sum_{i=0}^{N} x_i^2}.
$$

As illustrated for $N = 2$ in Figure 3, there is a narrow optimal reward region at the center, surrounded by a wide zero-reward area. The edges also offer suboptimal rewards. This configuration makes it unlikely for decentralized agents to randomly converge on the optimal region, often resulting in agents getting stuck in the large suboptimal zones. This setup presents significant challenges for exploration and coordination. To assess these dynamics, agent positions are randomly initialized at the start of each 25-timestep episode. The combination of a narrow optimal reward zone and decentralized partial observability creates a challenging multi-agent exploration problem. Success in this environment requires effectively mitigating the RO issue to discover joint policies that maximize cumulative reward.

**Multiple Particle Environment:** The primary objective is for two disk-shaped agents, $D_1$ and $D_2$, with centers $x_1$ and $x_2$ and radius $r_a$, to simultaneously reach a disk-shaped target or moving prey with center $x_t$ and radius $r_t$. To foster coordination, we introduce a penalty for scenarios where only one agent is within a specific distance from the target. Specifically, we define a disk $D$ centered on the target with radius $r_D = r_t + \alpha$ and penalize agents if only one is within $D$. This setup is designed to illustrate the RO problem, where agents might prefer staying outside $D$ rather than risking being the only one inside it. The task difficulty increases as $\alpha$ decreases. For our experiments, we maintain a default value of $r_a = 0.15$, $r_t = 0.05$, and set $\alpha = 0.3$ for the Predator-Prey task and $\alpha = 0.2$ for other tasks. The reward function designed to reflect the RO problem is given by:

$$
r_{\text{CN}} = \begin{cases}
R_{in} & \text{if } D_i \cap D \neq \emptyset, \, i = 1, 2 \\
R_{out} & \text{if } (D_1 \cup D_2) \cap D = \emptyset \\
R_{out} - p & \text{otherwise.}
\end{cases}
$$

Table 4 outlines the specific reward parameters used across different settings, showcasing how the reward values are adjusted to enhance or mitigate the challenges posed by the RO problem. This table provides a clear overview of how penalties and rewards are configured to encourage effective coordination and exploration strategies among agents.

Table 4: RO reward design for different scenarios in MPE task

| Setting | $R_{out}$ | $R_{in}$ | $p$ |
|---|---|---|---|
| CN | $-3(r_D + r_a)$ | $-3 \min_{i=1,2} \|x_t - x_i\|$ | 0.2 |
| More Penalty | $-3(r_D + r_a)$ | $-3 \min_{i=1,2} \|x_t - x_i\|$ | 0.5 |
| HT (Target A) | $-3$ | 0 | 0.5 |
| HT (Target B) | $-3$ | $-2.5$ | 0 |
| HA | $-3(r_D + r_a)$ | $-3 \min_{i=1,2} \|x_t - x_i\|$ | 0.2 |
| Predator-Prey | $-3(r_D + r_a)$ | $-3 \min_{i=1,2} \|x_t - x_i\|$ | 0.5 |
| Sequential (without cargo) | $-6$ | $-3$ | 0.5 |
| Sequential (with cargo) | $-6$ | $-0.5$ | 0.5 |

## B  Proof of Theorems 4.1 and 4.2

We first prove a theorem and lemma before moving on to the proof of Theorem 4.1.

**Theorem B.1.** *Let $\mathcal{T}_{\hat{S}}$ be the operator defined by $\mathcal{T}_{\hat{S}} Q(s, a_i) = \max_{s' \in \hat{S}} (R(s, s') + \gamma \max_{a'_i} Q(s', a'_i))$, then $\mathcal{T}_{\hat{S}}$ is a contraction in sup-norm.*

*Proof.* Let $Q_1$ and $Q_2$ be two Q-functions, i.e., real-valued functions on $\mathcal{S} \times \mathcal{A}_i$. By definition of the operator $\mathcal{T}_{\hat{S}}$ and of the argmax, we can rewrite the sup-norm $\|\mathcal{T}_{\hat{S}} Q_1 - \mathcal{T}_{\hat{S}} Q_2\|_\infty$ as

$$\|\mathcal{T}_{\hat{S}} Q_1 - \mathcal{T}_{\hat{S}} Q_2\|_\infty = \max_{s,a_i} \left| \max_{s' \in \hat{S}} \left( R(s, s') + \gamma \max_{a'_i} Q_1(s', a'_i) \right) - \max_{s' \in \hat{S}} \left( R(s, s') + \gamma \max_{a'_i} Q_2(s', a'_i) \right) \right| \quad (10)$$

$$= \max_{s,a_i} \left| (R(s, s'_1) + \gamma \max_{a'_i} Q_1(s'_1, a'_i)) - (R(s, s'_2) + \gamma \max_{a'_i} Q_2(s'_2, a'_i)) \right|, \quad (11)$$

where $s'_j = \arg\max_{s' \in \hat{S}}(R(s, s') + \gamma \max_{a'_i} Q_j(s', a'_i))$, $j = 1, 2$. It follows that

$$\|\mathcal{T}_{\hat{S}} Q_1 - \mathcal{T}_{\hat{S}} Q_2\|_\infty = \max_{s,a_i} \left| R(s, s'_1) - R(s, s'_2) + \gamma \max_{a'_i} Q_1(s'_1, a'_i) - \gamma \max_{a'_i} Q_2(s'_2, a'_i)) \right| \quad (12)$$

$$\leq \max_{s,a_i} |R(s, s'_1) - R(s, s'_2)| + \gamma \max_{s,a_i} \left| \max_{a'_i} Q_1(s'_1, a'_i) - \max_{a'_i} Q_2(s'_2, a'_i)) \right| \quad (13)$$

$$\leq \gamma \max_{s,a_i} \left| \max_{a'_i} Q_1(s'_1, a'_i) - \max_{a'_i} Q_2(s'_2, a'_i)) \right| \quad (14)$$

$$\leq \gamma \max_{s,a'_i} |Q_1(s, a'_i) - Q_2(s, a'_i)| \quad (15)$$

$$= \gamma \|Q_1 - Q_2\|_\infty, \quad (16)$$

which concludes the proof of the theorem. $\square$

Therefore, according to the contraction mapping theorem, $\mathcal{T}_{\hat{S}}$ converges to a unique fixed point which we denote by $Q$. Additionally, we denote by $Q^*$ the optimal Q-function, which is the fixed point of $\mathcal{T}_{\mathcal{S}}$.

We consider the following assumptions:

A.1  $R(s, \cdot)$ is Lipschitz continuous for any $s \in \mathcal{S}$

A.2  $Q^*$ is twice continuously differentiable function and Lipschitz continuous

A.3  For any $s \in \mathcal{S}$ and any maximizer $a^*$ of $Q^*(s, \cdot)$, it holds that $a^*$ is a maximizer of order 2, i.e., there are constants $c$ and $\delta$ such that

$$Q^*(s, a^*) - Q^*(s, a) \geq c\|a^* - a\|, \qquad \forall a \in \mathcal{A}_i, \|a^* - a\| \leq \delta.$$

**Lemma B.2.** *Define $q^*(s, s') = R(s, s') + \gamma \max_{a'} Q^*(s', a'_i)$ and consider Assumptions A.1-A.3. If $d(s'^*, \hat{s}'^*) \leq \epsilon$, then there exists a constant $K$ such that $|q^*(s, s'^*) - q^*(s, \hat{s}'^*)| \leq K\epsilon$ and where $s'^*, \hat{s}'^*$ are the best next state in $\mathcal{S}_{s,a_i}$ and $\hat{S}$, respectively.*

*Proof.* Based on the definition of $q^*$ and on the triangle inequality, it holds that

$$
\begin{aligned}
|q^*(s, s'^*) - q^*(s, \hat{s}'^*)| &= |R(s, s'^*) + \gamma \max_{a_i'} Q^*(s'^*, a_i') - R(s, \hat{s}'^*) - \gamma \max_{a_i'} Q^*(\hat{s}'^*, a_i')| \\
&\leq |R(s, s'^*) - R(s, \hat{s}'^*)| + \gamma |\max_{a_i'} Q^*(s'^*, a_i') - \max_{a_i'} Q^*(\hat{s}'^*, a_i')| \\
&\leq \epsilon K_R + \gamma |\max_{a_i'} Q^*(s'^*, a_i') - \max_{a_i'} Q^*(\hat{s}'^*, a_i')|,
\end{aligned}
\tag{17}
$$

with $K_R$ the Lipschitz constant for $R(s, \cdot)$. Let $a_i'^* = \arg\max_{a_i'} Q^*(s'^*, a_i')$ and $\hat{a}_i'^* = \arg\max_{a_i'} Q^*(\hat{s}'^*, a_i')$. According to Theorem 6.2 in (Still, 2018), there are $\epsilon, L > 0$ such that for all $s' \in \mathcal{S}$ verifying $\|s' - s'^*\| \leq \epsilon$, there exists a local maximizer $\tilde{a}_i'$ verifying

$$
\|\tilde{a}_i' - a_i'^*\| \leq L\|s' - s'^*\|.
$$

It follows that $\|(s'^*, a_i'^*) - (\hat{s}'^*, \hat{a}_i'^*)\|^2 \leq \epsilon^2(1 + L^2)$, and we finally obtain that

$$
|q^*(s, s'^*) - q^*(s, \hat{s}'^*)| \leq \epsilon K_R + \epsilon\sqrt{1 + L^2} K_Q = K\epsilon,
$$

with $K_Q$ the Lipschitz constant for $Q$ and with $K = K_R + \sqrt{1 + L^2} K_Q$. □

*Proof of Theorem 4.1.* We now need to consider the accumulation of error between time steps and therefore introduce an additional subscript $t$ on all the time-varying quantities. In addition, let $q(s, s') = R(s, s') + \gamma \max_{a_i'} Q(s', a_i')$ be the function related to the approximate Q-function $Q$ and define $\Delta_{t+1} = |q^*(s_t, \hat{s}_t'^*) - q(s_t, \hat{s}_t'^*)|$. Then, we have

$$
\begin{aligned}
|q^*(s_t, s_t'^*) - q(s_t, \hat{s}_t'^*)| &= |q^*(s_t, s_t'^*) - q^*(s_t, \hat{s}_t'^*) + q^*(s_t, \hat{s}_t'^*) - q(s_t, \hat{s}_t'^*)| \\
&\leq K\epsilon + \Delta_{t+1}.
\end{aligned}
\tag{18}
$$

We can manipulate the expression of $\Delta_{t+1}$ to make the left hand side of (18) appear, but at time $t + 1$. First, we observe that

$$
\begin{aligned}
\Delta_{t+1} &= |q^*(s_t, \hat{s}_t'^*) - q(s_t, \hat{s}_t'^*)| \\
&= |R(s_t, \hat{s}_t'^*) + \gamma \max_{a_{i,t+1}'} Q^*(\hat{s}_t'^*, a_{i,t+1}') - R(s_t, \hat{s}_t'^*) - \gamma \max_{a_{i,t+1}'} Q(\hat{s}_t'^*, a_{i,t+1}')| \\
&= \gamma \left| \max_{a_{i,t+1}'} Q^*(\hat{s}_t'^*, a_{i,t+1}') - \max_{a_{i,t+1}'} Q(\hat{s}_t'^*, a_{i,t+1}') \right|.
\end{aligned}
$$

The first term will have the optimal action at state $\hat{s}_t'^*$ as maximizer, and we denote $\hat{s}_{t+1}'^*$ the induced state. Similarly, we denote by $s_{t+1}'^*$ the state following from the optimal action under $Q$. It follows that

$$
\begin{aligned}
\Delta_{t+1} &= \gamma \left| R(\hat{s}_t'^*, \hat{s}_{t+1}'^*) + \gamma \max_{a_{i,t+2}'} Q^*(\hat{s}_{t+1}'^*, a_{i,t+2}') - R(\hat{s}_t'^*, s_{t+1}'^*) - \gamma \max_{a_{i,t+2}'} Q(s_{t+1}'^*, a_{i,t+2}') \right| \\
&= \gamma |q^*(\hat{s}_t'^*, s_{t+1}'^*) - q(\hat{s}_t'^*, \hat{s}_{t+1}'^*)| \\
&\leq \gamma K\epsilon + \gamma \Delta_{t+2}.
\end{aligned}
\tag{19}
$$

Based on (18) and (19), we have:

$$
\left| q^*(s_t, s_t'^*) - q(s_t, \hat{s}_t'^*) \right| \leq (1 + \gamma^2 + \cdots + \gamma^{T-1}) K\epsilon + \gamma^{T-1} \Delta_{t+T}
$$

As $T$ goes to the infinity, we have

$$
\left| q^*(s_t, s_t'^*) - q(s_t, \hat{s}_t'^*) \right| = \frac{1}{1 - \gamma} K\epsilon
$$

Since it holds that $|Q^*(s_t, a_{i,t}) - Q(s_t, a_{i,t})| = |q^*(s_t, s_{t+1}^*) - q(s_t, \hat{s}_{t+1}'^*)|$, we have

$$
|Q^*(s_t, a_{i,t}) - Q(s_t, \hat{a}_{i,t})| \leq \frac{1}{1 - \gamma} K\epsilon
$$

which concludes the proof of the theorem. □

*Proof of Theorem 4.2.* Consider the random variables $Y_k = |s'_k - \tilde{s}'^*|$ for all $k \in \{1, \ldots, M\}$. For simplicity, we used $c$ to denote $\tilde{s}'^*$ in the following. Their survival function is found to be

$$
\begin{aligned}
S_Y(y) &= P(|s'_k - c| > y) \\
&= P(s'_k > c + y) + P(s'_k < c - y) \\
&= \begin{cases}
1, & y \leq 0 \\
1 - \dfrac{y}{u}, & 0 < y \leq u - |c| \\
\dfrac{u + |c| - y}{2u}, & u - |c| < y < u + |c| \\
0, & y \geq u + |c|
\end{cases}
\end{aligned}
$$

The survival function of $\min_{k=1,\cdots,M} Y_k$ is $S_Y(y)^M$ and

$$
\begin{aligned}
\mathbb{E}\left[\min_{k=1,\ldots,M} Y_k\right] &= \int_0^u S_Y(y)^M dy \\
&= \int_0^{u-|c|} \left(1 - \frac{y}{u}\right)^M dy + \int_{u-|c|}^{u+|c|} \left(\frac{u + |c| - y}{2u}\right)^M dy \\
&= \frac{-u}{M+1} \left(1 - \frac{y}{u}\right)^{M+1} \bigg|_0^{u-|c|} + \frac{-2u}{M+1} \left(\frac{u + |c| - y}{2u}\right)^{M+1} \bigg|_{u-|c|}^{u+|c|} \\
&= \frac{u}{M+1} \left(1 + \frac{|c|^{M+1}}{u^{M+1}}\right) \\
&< \frac{2u}{M+1},
\end{aligned}
$$

which concludes the proof of the theorem. $\qquad\square$

## C  Ablation Study

### C.1  Negative Reward Shifting

To enhance exploration and mitigate overestimation in our MMQ approach, we have integrated a negative linear reward shifting technique (Sun et al., 2022). This method involves a downward adjustment of the reward function (by subtracting a constant from all rewards) during training. Analogous to starting with a higher initial value in the neural value function, this adjustment assigns inflated Q-values to less frequently visited state-action pairs, thus prioritizing them during the learning process. Our implementation of this insight prioritizes optimal next states, which are initially less common than suboptimal ones. By initializing the value function at a higher level, these optimal states receive more favorable evaluations. This technique proves particularly beneficial in our double-max-style updates, as it aids in the accurate selection of next states for updates and counteracts the overestimation of Q-values associated with chosen actions.

We conducted an ablation study on negative reward shifting in the differential games with $N = 2$ and $N = 4$, depicted in Figure 6. The results indicated that negative reward shifting significantly enhanced the performance of our algorithm. Additionally, we visualized the evolution of the learned Q-values during the initial 16 updates of training in Figure 7. With the same Q-value initialization, the central part of the state space, which represents infrequently achieved states, exhibited higher values than the edge parts under the negative reward shifting regimen. In contrast, when utilizing the standard positive reward setting, the central part displayed lower values compared to the edges initially, underscoring the effectiveness of this reward adjustment strategy in promoting a more balanced exploration and accurate value estimation.

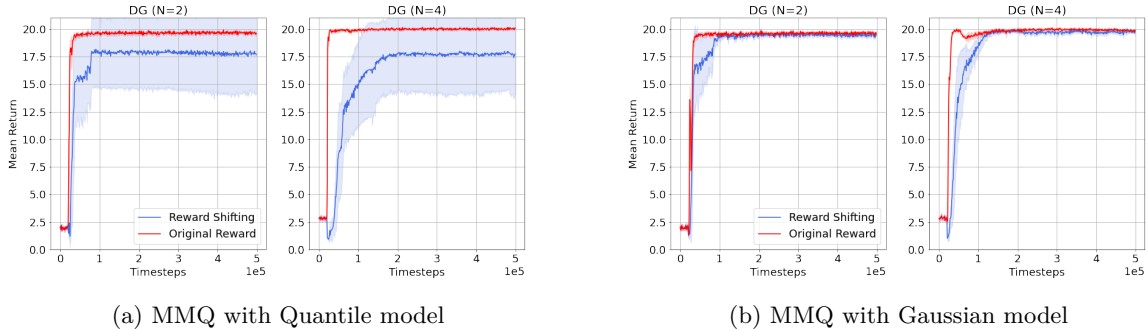

(a) MMQ with Quantile model          (b) MMQ with Gaussian model

Figure 6: Performance comparison between reward shifting and original reward

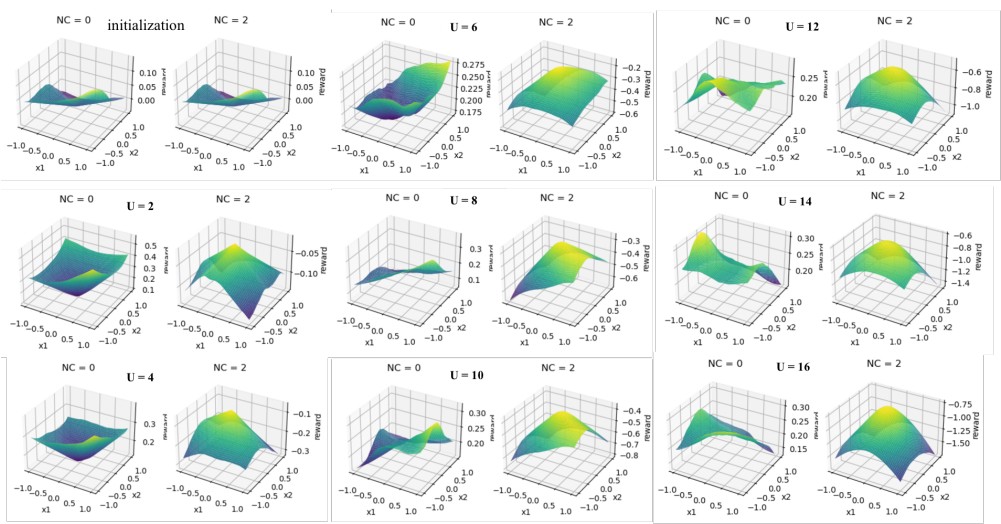

Figure 7: Comparison of Q-value for our algorithm with and without Negative reward shifting; NC=0, without negative reward shifting; NC=2, with negative reward shifting and the shifting constant = 2.

## C.2 Ensemble

In the MMQ algorithm, each agent uses two quantile models to capture variations in environment dynamics. While a single model on each side of the quantile bound provides valuable insights, its capacity to fully encapsulate the range of possible outcomes for state-action pairs can be limited. This limitation stems from the evolving nature of the agents' policies and the inherent uncertainties within the environment.

To enhance the robustness of our model and better address these challenges, we experimented with an ensemble approach. We utilized $K$ pairs of models, denoted as $\{g_{i,k}^{\tau_l}(s'|s, a_i; \phi_{k,i}^l)\}_{k=1}^{K}$ and $\{g_{i,k}^{\tau_u}(s'|s, a_i; \phi_{k,i}^u)\}_{k=1}^{K}$, where each model in the ensemble is characterized by its own unique parameter set $\phi_{k,i}$. This ensemble configuration allows for a broader representation of the dynamics, providing a more diverse set of potential outcomes. We define $\mathcal{S}_{s,a_i}^*$ as the union of outcomes from all models: $\mathcal{S}_{s,a_i}^* = \bigcup_{k=1}^{K} \{s'_{k,1}, \ldots, s'_{k,M}\}$, where $s'_{k,1}, \ldots, s'_{k,M}$ are independent and identically distributed samples drawn from the bounds predicted by each pair of quantile models.

During the initial learning phases, these $K$ models yield a broader range of predictions, as depicted in Fig.8b, reflecting different potential outcomes for the same state-action pairs. However, this approach did not lead to further improvement in performance, as a single quantile network was already sufficient to predict the bounds accurately. Consequently, this ensemble method did not enhance our algorithm's effectiveness in practice.

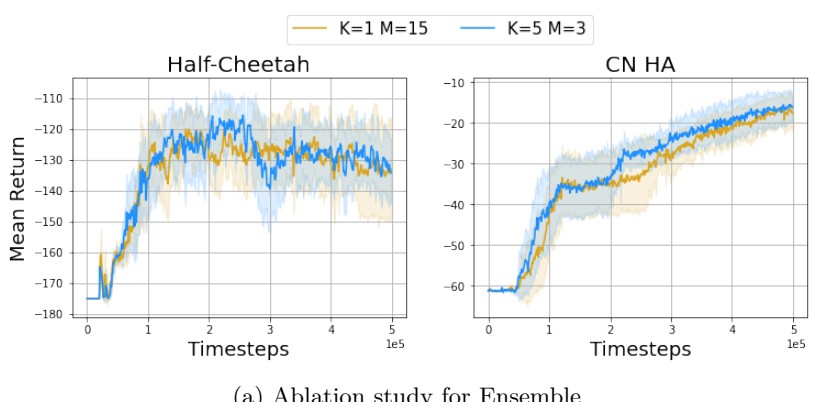

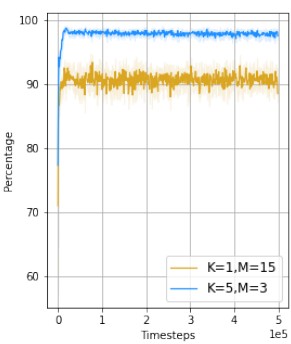

(a) Ablation study for Ensemble

(b) Percentage of true next states fall within the predicted quantile bound (each dim)

### C.3 Comparison between quantile model and Gaussian model

For the Gaussian model, the neural network parameters, are learned by minimizing a loss function based on the negative log-likelihood of the Gaussian distribution using experiences from its replay buffer $\mathcal{D}_i$:

$$L(\phi_i) = -\mathbb{E}_{s,a_i \sim \mathcal{D}_i} \left[ \sum_{d=1}^{D} \log \left( \mathcal{N}(s'_d | \mu_{\phi_i,d}(s,a_i), \sigma^2_{\phi_i,d}(s,a_i)) \right) \right]. \tag{20}$$

Here, $\mu_{\phi_i,d}(s,a_i)$ and $\sigma^2_{\phi_i,d}(s,a_i)$ are the neural network's predictions of the mean and variance for each dimension $d$ of the next state $s'$. The loss function sums the log-likelihoods across all dimensions $D$ of the state space, focusing on accurately modeling the distribution of each state dimension. As shown in 9b, the Gaussian model could also capture the other agents' influence on the observed state changes. The result in 9a showed MMQ with the quantile model is slightly better than MMQ with the Gaussian model.

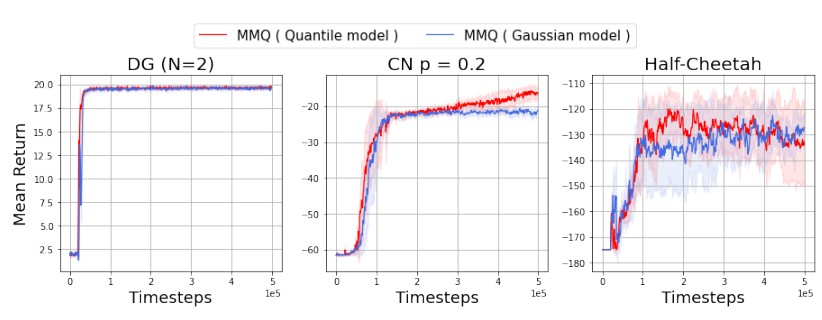

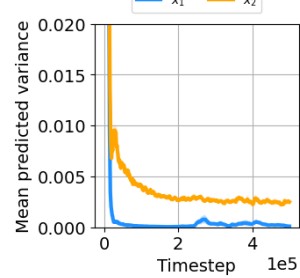

(a) Comparison between Gaussian model and Quantile model

(b) Predicted Variance of Agent 1's Forward Model in Differential Games DG (N=2)

## D Additional results

### D.1 Learning curves in stochastic environment

To assess the robustness of our algorithm in the stochastic environment, we add Gaussian noise to the transition and the reward function. For stochastic state transition, we add a Gaussian noise to the position update of each agent: $x_i = \text{clip}\{x_i + 0.1 \times a_i + z, -1, 1\}$, while $z \sim \mathcal{N}(0, \sigma^2)$. Here we test in two levels of stochasticity: $\sigma_s = 0.02$ and $\sigma_s = 0.05$. For the stochastic reward, we add Gaussian noise to the reward, $z \sim \mathcal{N}(0, \sigma^2)$, and the reward becomes $r = r + z$. We also test in two levels of noise: $\sigma_r = 0.05$ and $\sigma_r = 0.1$. In Figure 10, with noise in state transition, our algorithm still perform the best compared to other baselines.

Although we use a deterministic function as reward function, our algorithm still find the global optimum for all seeds when there exits reward noise whereas the performance of other baselines degraded.

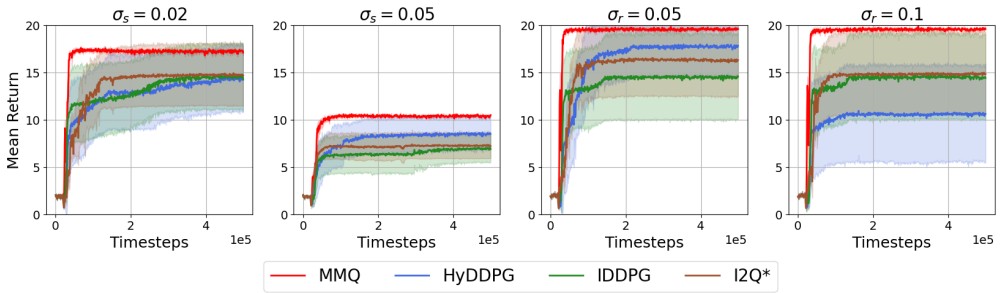

Figure 10: Learning curves in stochastic version of differential game (N=2)

## D.2 Default Reward setting

We test our algorithm in cooperative navigation and Predator-Prey with the default reward setting (Lowe et al., 2017). In the cooperative navigation, the result showed with a learned reward function, our algorithm learned a bit slower than two model-free baselines, HyDDPG and IDDPG, but finally achieved a similar level. When using the true reward from experiences (we then use $r(s')$ rather than $r(\hat{s}')$), our algorithm could achieve a similar performance as two baselines. Therefore, we assumed the reward estimation error in dense reward settings may impact our algorithm's performance slightly. In Predator-Prey with default sparse reward that agents would get +10 only when they collided with the prey, our algorithm outperforms all baselines.

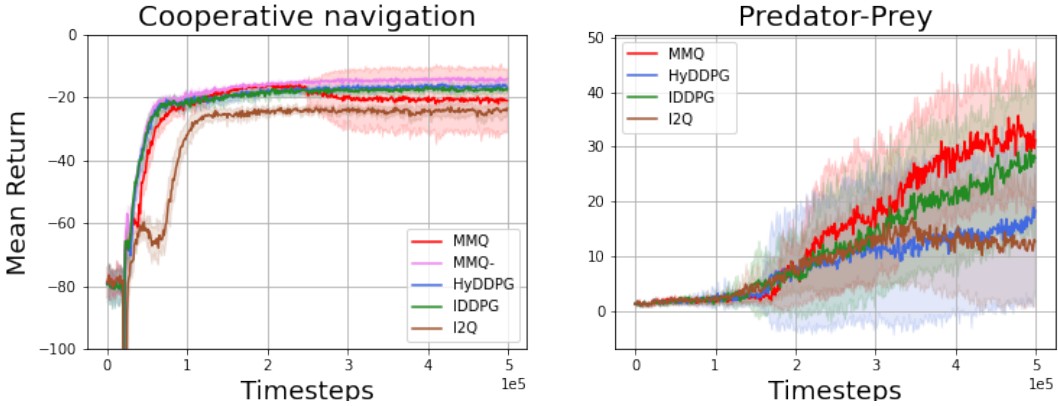

Figure 11: Cooperative navigation and Predator-Prey environment with default reward; MMQ$^-$ represent that we replace the learned reward function using the true reward from the experience;

## D.3 Additional Results for MAmujoco

We add results for three scenarios in Figure 12: (1) Half-Cheetah 4|2, there are two agents, one agent control 4 joints and the other agent controls 2 joints. (2) Ant $2 \times 4$, two agents, each agent control 4 joints. (3) Ant $4 \times 2$, four agents and each agent control 2 joints. MMQ slightly outperforms all other baselines.

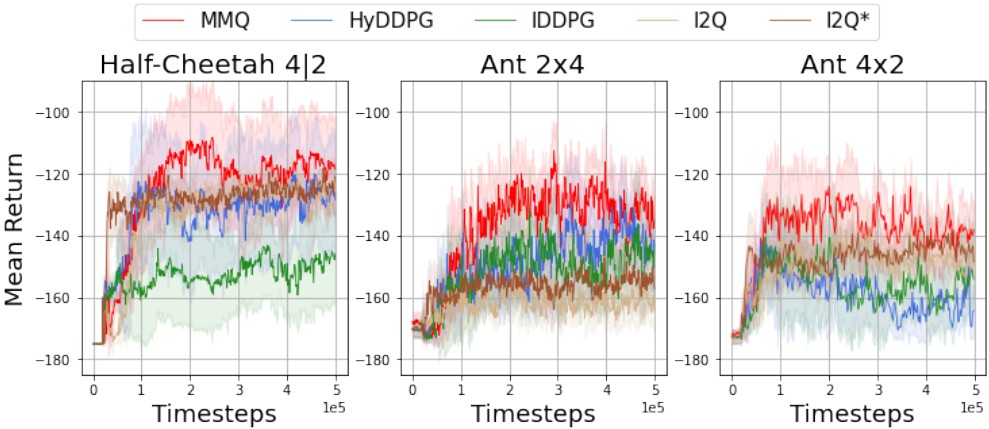

Figure 12: Learning curves on Multi-Agent Mujoco with RO reward setting

## E   Hyperparameters

Table 5: Common Model and training hyperparameters used in all algorithms

|  | Parameter |
| --- | --- |
| Model architecture | FC layers[256,256] |
| Forward model architecture | FC layers[256,256] |
| Replay Buffer size | 550000 |
| Batch size | 100 |
| Optimizer | Adam |
| Learning rate | 0.001 |
| Discount factor gamma | 0.99 |
| Network initialization | Xavier |
| Activation | ReLU |
| Number of seeds | 8 |
| Total environment step | 500000 |
| Exploration epsilon | 0.1 |
| Episode length | 25 Steps (expect 50 Steps for MPE sequential task) |
| Experience collection steps before training $T_{pretrain}$ | 20000 |

The hyperparameters used in our experiments are listed in Table 5. All experiments were conducted on an NVIDIA TITAN V GPU. The training for each task typically completed within 5 hours.