# OpenReview forum: "Mitigating Relative Over-Generalization in Multi-Agent Reinforcement Learning"
_TMLR — Accepted by TMLR_

### Review · Reviewer_PxMX · 2024-08-03

**Summary Of Contributions:**

The paper introduces MaxMax Q-Learning (MMQ), an algorithm designed to mitigate RO by sampling and evaluating potential state transitions and selecting those with maximal Q-values. This process aims to refine approximations of ideal state transitions and align more closely with optimal joint policies.

**Audience:**

Yes

**Broader Impact Concerns:**

Broader Impact is stated in this paper.

**Claims And Evidence:**

Yes

**Requested Changes:**

Please see the questions in the previous session.

**Strengths And Weaknesses:**

Strengths:

MMQ introduces a new approach to address the RO problem, enhancing convergence and sample efficiency.

The paper provides both theoretical analysis and empirical evaluations demonstrating MMQ's effectiveness across various environments.

The experiments results are good and adequate to show the effectiveness of the paper.

Questions:

In my understanding, to get the maximum expected return will also face a scalability problem, how did you solve that?

How does the computational complexity of MaxMax Q-Learning (MMQ) compare to traditional Q-learning in large-scale multi-agent environments, and what are the specific computational trade-offs?

---

> ### Author Response · Authors · 2024-09-04
>
> Thank you for your insightful questions regarding the scalability and computational complexity of our MaxMax Q-Learning (MMQ) approach. Let me address each point:
>
> **Scalability in obtaining maximum expected return:**
>
> You're correct that obtaining the maximum expected return can face scalability challenges, especially in multi-agent environments. Our approach addresses this through:
>
> a) Iterative sampling and evaluation: We use an iterative process of sampling, evaluating, and selecting optimal states during learning. This allows us to approximate the maximum expected return without exhaustively searching the entire state space.
>
> b) Decentralised learning: By having each agent sample from its own experiences rather than depending on information from all agents, we significantly improve scalability compared to centralised training or communication-heavy frameworks.
>
> c) Adaptive sampling: As shown in our ablation study (Section 5.3), even a small number of samples (e.g., 15) can lead to good performance. The number of samples can be adjusted based on the complexity of the environment and computational resources available.
>
> d) Quantile models: Our use of quantile models helps efficiently estimate the range of possible next states, contributing to scalability by focusing on the most relevant parts of the state space.
>
> e) Balance of exploration and exploitation: The adaptive sampling approach allows for a balance between exploration and exploitation, which is crucial for scalability in complex environments.
>
> **Computational complexity compared to traditional Q-learning:**
>
> While our approach does introduce some additional computational overhead compared to traditional Q-learning, the trade-offs are favourable:
>
> a) Runtime analysis: Our experiments all involve continuous action spaces. Therefore, we used IDDPG instead of IQL. We provide an analysis of the runtime and memory footprint of our method compared to all the baselines (see table below). The results show that MMQ's runtime was approximately 30% longer than IDDPG's for N=2 agents, and about 32% longer for N=5 agents.
>
> b) Memory footprint: Our method requires additional memory for the quantile models and sampled states. However, as shown in the table, MMQ used only slightly more GPU memory than IDDPG (less than 1% increase), demonstrating its memory efficiency.
>
> c) Scalability with increasing agents: The computational complexity of MMQ scales linearly with the number of agents, similar to traditional Q-learning. However, MMQ shows better scalability in terms of learning performance, as demonstrated in our results with varying numbers of agents (Tables 2 and 3 in the paper).
>
> d) Sample efficiency: While MMQ requires more computation per step, it often requires fewer total steps to converge to an optimal policy, especially in environments prone to relative over-generalisation. This can lead to overall computational savings in many scenarios.
>
> e) Performance trade-off: The additional computational cost of MMQ is offset by its superior performance in terms of convergence speed and final policy quality, particularly in challenging multi-agent environments.
>
> To illustrate these points, we've prepared a table comparing the runtime and memory usage of MMQ against our baselines across different experimental settings in the Differential Game environment:
>
> | **Differential Game** | **Runtime (s/1e4 steps)** | | **GPU Mem. (MiB)** | |
> |-----------------------|--------------------------|---------------------|------------------|-----------------|
> |                       | N=2                      | N=5                 | N=2              | N=5             |
> | **IDDPG**             | 37.6                     | 88.65               | 1065             | 1073            |
> | **HyDDPG**            | 45.0                     | 116.02              | 1065             | 1073            |
> | **I2Q**               | 41.1                     | 99.14               | 1071             | 1087            |
> | **I2Q***              | 103.2                    | 260.20              | 1071             | 1087            |
> | **MMQ(K=1,M=15)**     | 49.2                     | 117.13              | 1069             | 1079            |
>
> As shown in the table, while MMQ does introduce some additional computational overhead, it remains competitive in terms of both runtime and memory usage, especially considering its performance benefits. Notably, MMQ's memory usage is comparable to simpler methods like IDDPG and HyDDPG, and it's more efficient in this regard compared to I2Q and I2Q*.
>
> **Limitations and future work:**
>
> It's important to note that while MMQ shows promising results, there may be scenarios where the additional computation might not be justified, particularly in very simple environments where traditional methods perform adequately. Additionally, as the state-action space grows extremely large, the sampling process might require further optimisation.

---

> ### Author Response · Authors · 2024-09-04
>
> Future work will explore potential optimisations, such as parallel sampling and more efficient quantile model architectures, to further improve the computational efficiency of MMQ.
>
> In summary, while MMQ does introduce some additional computational complexity, its scalability and performance benefits, particularly in challenging multi-agent environments prone to relative over-generalisation, outweigh these costs in many scenarios. The method provides a favourable trade-off between computational requirements and improved learning outcomes.

---

### Review · Reviewer_VGaS · 2024-08-19

**Summary Of Contributions:**

The paper introduces MaxMax Q-Learning (MMQ), a novel method designed to mitigate the issue of relative over-generalization (RO) in decentralized multi-agent reinforcement learning. MMQ addresses RO by iteratively sampling and evaluating potential next states, selecting those with the maximal Q-values.
MMQ refines approximations of ideal state transitions. This work provides a theoretical analysis demonstrating MMQ’s potential to converge to globally optimal joint policies under certain assumptions.
Empirical evaluations across various environments prone to RO show that MMQ demonstrates stronger overall performance than existing baselines.

**Audience:**

Yes

**Claims And Evidence:**

Yes

**Requested Changes:**

Please see the weaknesses above.

**Strengths And Weaknesses:**

## Strength
This paper is well-structured, presenting clear motivations and solution methods. This work provides detailed theoretical results and illustrative visualizations.

## Weaknesses
### Methodology

In Eq.6, there is no stop-gradient operation applied to the target value?

### Experiment
I encourage the authors to perform more comprehensive experiments on the complex MA-Mujoco benchmark. The work only includes the Half-Cheetah 2x3 task. More tasks such as Half-Cheetah 6x1, and Ant 2x4/4x2 are also suitable evaluation scenarios by implementing the RO reward in a similar way.

MMQ demonstrates the best overall performance, yet it only outperforms all baselines in two tasks. This also necessitates more experiments in MAMujoco tasks to better support their claims.

### Writing
Sec 1, "...may encounter scalability and privacy issues". CTDE methods do not necessarily gather local information from all agents. Some approaches could directly utilize global state information, thereby circumventing the scalability issue. Also, whether the privacy issue is considered in practice depends on the task. Task completion in many domains is not concerned with privacy.

I suggest the authors motivate decentralized learning with another example. The mixed autonomous traffic example does not necessitate gathering information or communicating with *all* agents. In practice an agent only needs to focus on its surroundings and ignore others.
It is worth noting that CTDE and communication allow information exchange, but it does not mean every CTDE/communication method must utilize information from the entire population.

Sec 2. in the first paragraph, the weighted QMIX work is cited as the QMIX paper.

Sec 3.1, typo in the individual dynamics equation: $\mathbf{a}-i$ should be  $\mathbf{a}_{-i}$ under the summation symbol.

Sec 4.1, on page 5, first sentence of the second paragraph repeats the last sentence of the first paragraph.

---

> ### Author Response · Authors · 2024-09-04
>
> Thank you for your thorough review and constructive feedback on our paper. We greatly appreciate your insights and the time you've taken to provide detailed comments. Below, we address each of your points.
>
> We're pleased that you found our paper well-structured, with clear motivations, solution methods, theoretical results, and illustrative visualisations. We aimed to present our work as clearly and comprehensively as possible.
>
> **Methodology:**
>
> Regarding the absence of a stop-gradient operation in Eq. 6, we apologise for the lack of clarity in our original manuscript. We did indeed apply a stop-gradient to the target value to prevent gradient flow back to the next state estimation process. This detail was inadvertently omitted in our presentation. We have explicitly indicated this in the revised version to ensure clarity. The stop-gradient operation is important for the stability of the algorithm as it prevents the target network from being directly influenced by the learning process of the next state estimation, which could lead to instability or divergence.
>
> **Experiments:**
>
> We appreciate your suggestion to conduct more comprehensive experiments on the MA-Mujoco benchmark. In response:
>
> - We have expanded our experiments to include the Half-Cheetah 4|2 task (where one agent controls 4 joints and the other controls 2 joints), Ant 2x4 and Ant 4x2 task in addition to the original Half-Cheetah 2x3 task. The results of these new experiments are now included in the revised manuscript (see Figure 12 in Appendix D.3 and the accompanying discussion).
> - We acknowledge the value of testing on scenarios with more partitions like Half-Cheetah 6x1. However, in this scenario, each agent only controls one joint, resulting in a very limited observation range. This limitation makes it challenging for our algorithm and all baselines to learn strong policies. Our current additional experiments with different configurations for Half-Cheetah and Ant provide insights into MMQ's performance across various multi-agent scenarios with different levels of complexity and agent cooperation.
> - We recognise that our current results show MMQ outperforming all baselines in only some tasks. The additional experiments help to further substantiate our claims about MMQ's effectiveness, particularly in environments prone to relative over-generalisation. These new experiments showcase MMQ's performance in scenarios with different levels of complexity and agent cooperation, allowing us to better understand the algorithm's strengths and limitations. We have updated our discussion based on these new results, providing a more nuanced analysis of MMQ's performance across a broader range of scenarios.
>
> **Writing:**
>
> 1. Regarding the CTDE discussion in Sec 1, we thank you for highlighting the nuances we overlooked. You're correct that not all CTDE methods necessarily gather information from all agents, and that global state information can sometimes be used directly. We have revised this section to more accurately represent the range of CTDE approaches and their implications for scalability. Our revised text now provides a more balanced view, acknowledging the diversity of CTDE methods and clarifying that privacy concerns are context-dependent and not universally applicable.
> 2. We appreciate your suggestion to provide a different example for motivating decentralised learning. We agree that the mixed autonomous traffic example has limitations. We have replaced it with a more suitable scenario of large-scale drone swarms for tasks such as search and rescue or surveillance. This example better illustrates the need for fully decentralised learning, as centralised training or extensive communication can be impractical due to network delays, dynamic environments, and potential communication failures.
> 3. Thank you for catching the citation error in Sec 2. We have corrected the reference to accurately cite the weighted QMIX work instead of the original QMIX paper.
> 4. The typo in the individual dynamics equation in Sec 3.1 has been fixed, with $a_{-i}$ now correctly placed under the summation symbol.
> 5. We have removed the repetition in the first two paragraphs of Sec 4.1 to improve clarity and conciseness.

---

### Review · Reviewer_sjZT · 2024-08-25

**Summary Of Contributions:**

This paper studies multi-agent reinforcement learning in a decentralized setting. Without explicit coordination, each agent may select actions based on how they expect the others to act, which may not be collectively optimal and can lead to relative over-generalization (RO). To address this challenge, the paper proposes a mechanism that introduces implicit coordination by guiding the agents towards the optimal joint policy, utilizing Q-learning, Monte Carlo optimization, and quantile models. The paper gives some theoretical justification towards understanding why this approach may work, and then provides an empirical evaluation in three environments that are designed to induce RO problems.

**Audience:**

Yes

**Broader Impact Concerns:**

Nothing specific.

**Claims And Evidence:**

Yes

**Requested Changes:**

I've listed some suggestions above. In addition:
- How should one determine the duration of each phase, specifically $T_\text{pretrain}$?
- The discussion of "non-stationarity" in Section 1 can be a bit confusing, as it may suggest that $P_\text{env}$, instead of $P_i$, changes over time. This only becomes clearer to me after the problem setting is introduced in Section 3.
- There are some typos and grammar mistakes in Section 3.1.
- The caption of Figure 1 is confusing -- what does the yellow agent’s "position" mean? What does it mean for the blue agent to "select" samples?
- What aspects of Theorem 4.1 are specific to the multi-agent setting? Can it be applied to single-agent RL?
- For Theorem 4.2, would there be an exponential dependence if the states are in a higher-dimensional Euclidean space?
- It is mentioned in Section 1 that the theoretical results assume "perfect knowledge of forward and value functions." Can you elaborate on this?
- In Equation (5), what is the definition of $D_i$?
- In the experiments, it would be nice to include a formal definition of "return".
- Why is the approach called MaxMax?
- Can you further discuss the setting and results in (Jiang & Lu, 2022) and how the proposed approach in this work differs?

**Strengths And Weaknesses:**

Strengths:
- I want to preface this by acknowledging that I am not an expert on multi-agent reinforcement learning, but the studied problem and the challenges in RO and non-stationarity (from each agent's perspective) seem interesting, well-motivated, and relevant to the community.
- The proposed mechanism integrates new techniques from sampling and uncertainty quantification into the existing approach of implicit coordination through modeling ideal transition probabilities.
- The theoretical results offer insights into how accurately estimating the best next state is connected to learning the optimal policy.
- An empirical study in three different environments shows superior performance in comparison with three benchmark algorithms.

Weaknesses:
- Despite the authors' efforts to explain the intuitions behind the methodology, the paper can be somewhat difficult to follow in parts. For example, it is unclear what exactly the agents can communicate in this decentralized setting -- do they observe the actions of other agents? Additionally, Figure 1 mentions a replay buffer, but this is not discussed elsewhere in the paper. Furthermore, the pseudocode (with details such as the use of $\epsilon$-greedy) is deferred to the appendix -- while I can generally grasp how the approach works, it can be hard to fully understand where each part fits into the mechanism; it would be helpful to include a high-level description of the algorithm in the form of pseudocode in the main paper.
- The theoretical analysis is a nice addition, but it does not seem to fully demonstrate convergence to the optimal joint policy. In particular, can one formally show how (fast) $\hat{\mathcal{S}}_{s,a_i}$ shrinks and zeros in on $s'^{*}$?
- The experiments used environments with hand-crafted reward functions that may lead to RO. It would be nice to evaluate the approach in more realistic or real-world settings.

---

> ### Author Response · Authors · 2024-09-04
>
> Thank you for your thorough review and insightful comments. We appreciate your recognition of the relevance and motivation behind our work, particularly your acknowledgment of the challenges in RO and non-stationarity in multi-agent reinforcement learning. We're pleased that you found our integration of sampling and uncertainty quantification techniques into implicit coordination approaches noteworthy, and that our theoretical results and empirical study demonstrated value. We'll address each of your points below.
>
> **Clarifications on the decentralised setting and methodology:**
>
> 1. Fully decentralised setting: In our fully decentralised setting, there is no communication among agents. Each agent can access the state information (e.g., positions of agents and targets) but cannot observe the policies or actions of other agents. This leads to independent decision-making, which can result in the RO problem as agents cannot accurately evaluate their choices due to limited information about others' decisions.
> 2. Replay buffer in Figure 1: We apologise for the lack of clarity. Figure 1 illustrates the difference between our approach and independent Q-learning. Each agent perceives the positions of both agents, constituting the state information $s=(x_\text{blue},x_\text{yellow})$. Like independent Q-learning, our algorithm is an off-policy, value-based approach. We sample experiences $(s,a_i,r,s')$ from the replay buffer to update the Q-value. Since one agent has no information about the other agent's action, a series of experiences based on the same $(s,a_i)$ may lead to different next states $s'$, such as $s'^1,s'^2,s'^3,\cdots$, corresponding to the different positions in the figure.
> 3. Pseudocode: We have moved the pseudocode to the main paper and included a high-level description of the algorithm.
>
> **Theoretical analysis:**
>
> 1. Convergence: You're correct that our theoretical analysis doesn't fully demonstrate convergence to the optimal joint policy. $\hat{\mathcal{S}}_{s,a_i}$ is approximated by quantile bounds learned by neural networks, making it challenging to formally prove how fast it shrinks. Empirically, we show that the estimated quantile bound captures over 90% of true next states during learning (Figure 5(b)), suggesting a high probability that the best next state $s'^*$ is included in our estimated set.
>
> 2. Thank you for this insightful question about Theorem 4.1. While the theorem itself is not explicitly formulated in multi-agent terms, its relevance and application are particularly significant in the multi-agent context of our work. Here's why:
>
>     (1) Best next state concept: In our multi-agent setting, the "best next state" s'* implicitly includes the optimal actions of other agents. This concept is central to addressing the relative over-generalization (RO) problem in multi-agent systems, where agents need to coordinate their actions for optimal joint behavior.
>
>     (2) Non-stationarity: The theorem helps bound the error in Q-value estimation when the best next state is approximated. In multi-agent settings, this approximation is crucial due to the non-stationarity introduced by other agents' changing policies.
>
>     (3) State set estimation: The set Ŝ in the theorem represents our algorithm's estimation of possible next states, which in the multi-agent case, accounts for the uncertainty about other agents' actions.
>
>     (4) Convergence implications: In the multi-agent context, the theorem suggests that as our estimation of the best next state improves (considering other agents' optimal actions), our Q-values converge closer to the optimal values for the joint policy.
>
>     While the theorem could theoretically apply to single-agent settings with non-stationary environments, its practical significance is much higher in our multi-agent framework. It provides theoretical support for our approach to mitigating the RO problem by implicitly modeling optimal joint behavior through next state estimation.
>
> 3. Theorem 4.2 and higher-dimensional states: You're correct that there could be an exponential dependence as dimensionality increases. For a $d$-dimensional state space with $M$ samples, we can bound the expected minimum value for the maximum distance over dimensions as:
> $\mathbb{E}[\min_k \max_j X_{j,k}] \leq \int (1 - F(y)^d)^M dy$
> where $F$ is the CDF of $X_{j,k}$. However, in multi-agent scenarios, we typically consider partially-observed MDPs, which would limit the growth of dimension $d$.
> 4. Perfect knowledge assumption: This refers to (1) a perfect forward model (quantile model) to determine the bound for possible next states, and (2) a perfect value function to select the truly best next state from these possibilities.

---

> ### Author Response · Authors · 2024-09-04
>
> **Experimental details and clarifications:**
>
> 1. Duration of $T_\text{pretrain}$: We set this to 20,000 steps to collect initial experiences before training begins. Varying this number did not significantly impact performance.
> 2. Definition of $D_i$: $D_i$ is the individual replay buffer, including experiences $(s,a_i,r,s')$.
> 3. Formal definition of "return": We've added this at the start of Section 5.3 in the revised manuscript.
> 4. Meaning of "MaxMax": Our algorithm is named MaxMax Q-learning because there are two max operators in the Bellman update: the first over the estimated set $\hat{\mathcal{S}}$ for next possible states, and the second over the action for the Q-value $\max_{a'_i}Q(s',a'_i)$.
>
> **Comparison with Jiang & Lu(2022):**
>
> 1. Setting: While I2Q addressed non-stationarity, we focus on relative over-generalisation, a specific issue exacerbated by non-stationarity in multi-agent interactions.
> 2. Algorithm: Both approaches incorporate optimism in learning, but differ in key aspects. I2Q uses the $Q_{SS}(s,s')$ value function and a forward model to predict the best next state directly. Our algorithm starts from the uncertainty in multi-agent systems and employs an iterative approach to estimate a set of possible next states and approximate the best next state.
> 3. Results: Our approach enhances performance in environments characterised by RO problems compared to I2Q. Our experiments also show that with slight adjustments (training the critic more frequently than the actor), both IDDPG and Hysteretic DDPG perform significantly better.
>
> **Addressing specific requests:**
>
> 1. Non-stationarity discussion: We've revised this in Section 1 for clarity.
> 2. Typos and grammar in Section 3.1: We've carefully reviewed and corrected these.
> 3. Figure 1 caption: We've clarified that "yellow agent's position" refers to possible next states due to different policies of the yellow agent in the replay buffer, and "blue agent selects samples" means choosing which past experiences to use for updating its Q-value.
> 4. More realistic tasks: We appreciate your suggestion for more realistic or real-world settings. In response to this, we have added additional experiments, including different configurations of the Half-Cheetah task  and Ant task (2x4, 4x2). These environments and tasks are of comparable complexity to those used in other MARL publications that explore specific algorithmic shortcomings, which in our case is the RO problem.
> It's important to note that our main focus in this work has been on highlighting the RO problem and proposing an algorithm that can mitigate that issue in the particular setup we've described. The environments we've chosen, while seemingly simple, are designed to induce RO problems, allowing us to clearly demonstrate our algorithm's effectiveness in addressing this specific challenge.
> We acknowledge that the proposed algorithm is not yet ready to address real-world multi-agent tasks 'as is', but this is also the case for many other contemporary MARL algorithms at similar stages of development. Our work provides a foundation for understanding and addressing RO in multi-agent settings, which can inform future research aimed at more complex, real-world applications.
> In future work, we plan to extend our approach to more complex scenarios, building upon the insights gained from these focused experiments. This gradual progression from targeted, problem-specific environments to more realistic settings is a common and valuable approach in advancing MARL research.
>
> We appreciate your thorough review, which has helped us improve the clarity and rigour of our work. We hope these clarifications address your concerns and welcome any further questions.

---

### Decision · Action_Editor_kEMY · 2024-10-17

**Recommendation:** Accept with minor revision

**Comment:**

The paper presents an algorithm to address "relative over-generalization" (RO) caused by fully-decentralized actors. From the paper:

"Another critical issue in decentralized MARL is Relative Over-generalization (RO), where agents prefer suboptimal policies because individual actions seem preferable in the absence of coordinated strategies."

I find this to be a peculiar name for this phenomenon. Why is this about "generalization" at all -- where is the generality here? If anything, the agents are *overfitting* to their (incorrect) local assumptions. They're all independently solving the "wrong" problem. One could argue that MMQ is making them "more general" by considering the wider problem rather than the agents fixating on a local sub-problem, so this can be lead to confusion. If this is a new term being proposed in this paper, I would advise a careful reconsideration, or at least some justification in the text for why generalization is in the name.

Aside from this, the paper presents an algorithm that addresses the problem, both in theory and practice. The reviewers had a number of comments and criticisms regarding technical clarity which have been addressed in the new versions.

The reviewers remain skeptical about the limited experiments and scaling potential as the population grows. I recommend briefly describing the limitations in its own section or subsection to elaborate on any points or settings that could be problematic from a potential user's perspective.

**Audience:**

All of the official recommendations answered yes for target audience. I also agree with the recommendations.

**Claims And Evidence:**

All of the official recommendations answered yes for claims and evidence. I agree with the recommendations.

---

> ### Author Response · Authors · 2024-11-07
>
> Thank you for your insightful feedback on our paper. In response, we have made revisions aimed at clarifying the terminology and addressing the scalability concerns raised.
>
> Clarification of "Relative Over-generalization" (RO): We acknowledged that "Relative Over-generalization" (RO) is an established term in the literature, and we are not introducing it anew. In the revised text, we cite Wiegand (2004) as a foundational source and reference additional works that use this terminology. We also expanded our explanation to clarify that RO reflects agents’ tendency to adapt to limited interactions, focusing on exploratory behaviors without coordinated strategies. This overfitting to decentralized interactions often leads agents to prefer robust yet suboptimal solutions, which we now state more explicitly to avoid potential confusion with the term "generalization."
>
> Discussion of Limitations: To directly address your suggestion, we expanded the “Conclusions” section into a “Discussion and Conclusions” subsection, where we explicitly outline the limitations of our approach. This includes a discussion on potential challenges related to scalability, such as the exponential dependence in Monte Carlo optimization as state-space dimensionality grows and the implications of this in high-dimensional spaces. We also address potential scalability issues in environments with large numbers of agents, where increased agent populations can lead to computational costs and performance bottlenecks. Additionally, we outline possible future extensions that could reduce sample requirements and improve scalability, offering context for our algorithm’s applicability in larger, complex environments.
> We hope these revisions address your feedback fully, and we are grateful for the constructive input, which has been invaluable in refining our work.